



# Rapid and sustained environmental responses to global warming: The Paleocene–Eocene Thermal Maximum in the eastern North Sea

Ella W. Stokke[1], Morgan T. Jones[1], Lars Riber[2], Haflidi Haflidason[3,4], Ivar Midtkandal[2], Bo Pagh Schultz[5], and Henrik H. Svensen[1]

[1]CEED, University of Oslo, PO Box 1028, 0315 Oslo, Norway

[2]Department of Geosciences, University of Oslo, P.O. Box 1047, Blindern, NO 0316 Oslo, Norway

[3]Department of Earth Science, University of Bergen, Allégt. 41, N-5007 Bergen, Norway

[4]Bjerknes Centre for Climate Research, Jahnebakken 5, 5007 Bergen, Norway

[5]Museum Salling, Fur Museum, Nederby 28, 7884 Fur, Denmark

*Correspondence to*: Ella W. Stokke (e.w.stokke@geo.uio.no)

**Abstract**

The Paleocene–Eocene Thermal Maximum (PETM; ~55.9 Ma) was a period of rapid and sustained global warming associated with significant carbon emissions. It coincided with the North Atlantic opening and emplacement of the North Atlantic Igneous Province (NAIP), suggesting a possible causal relationship. Only a very limited number of PETM studies exist from the North Sea, despite its ideal position for tracking the impact of both changing climate and the NAIP explosive and effusive activity. Here we present sedimentological, mineralogical, and geochemical proxy data from Denmark in the eastern North Sea, exploring the environmental response to the PETM. An increase in the chemical index of alteration and a kaolinite content up to 50 % of the clay fraction indicate an influx of terrestrial input shortly after the PETM onset and during the recovery, likely due to an intensified hydrological cycle. The volcanically derived minerals zeolite and smectite comprise up to 36 % and 90 % of the bulk and clay mineralogy respectively, highlighting the NAIPs importance as a sediment source for the North Sea and in increasing the rate of silicate weathering during the PETM. XRF element core scans also reveal possible hitherto unknown NAIP ash deposition both prior to and during the PETM. Geochemical proxies show that an anoxic environment persisted during the PETM body, possibly reaching euxinic conditions in the upper half with high concentrations of Mo (>30 ppm), S (~4 wt%), and pyrite (~7 % of bulk), and low Th/U (<2 ppm). At the same time, export productivity and organic matter burial reached its maximum intensity. These new records reveal that negative feedback mechanisms including silicate weathering and organic carbon drawdown rapidly began to counteract the carbon cycle perturbations and temperature increase, and remained active throughout the PETM. This study highlights the importance of shelf sections in tracking the environmental response to the PETM climatic changes, and as carbon sinks driving the PETM recovery.





## 1. Introduction

The early Cenozoic was a period characterized by long-term warming, punctuated by transient periods of rapid global hyperthermal events (Zachos et al., 2008; Hollis et al., 2012; Cramwinckel et al., 2018). The most
pronounced of these periods was the Paleocene–Eocene Thermal Maximum (PETM; ~55.9 Ma; Kennett and Stott, 1991; Thomas and Shackleton, 1996; Westerhold et al., 2018), during which global surface temperatures rose rapidly by 4–5 °C (Dunkley Jones et al., 2013; Frieling et al., 2017). The PETM is associated with a large input of $^{12}$C-rich carbon to the ocean–atmosphere system resulting in a 2.5–8 ‰ negative carbon isotope excursion (CIE) in the terrestrial and marine sedimentary record (McInerney and Wing, 2011). The PETM CIE lasted up to 200
kyr (Westerhold et al., 2018), and is characterised by a rapid onset (~1–5 kyr; Kirtland-Turner et al., 2017), followed by a stable body (~100 kyr; van der Meulen et al., 2020) and a gradual recovery towards background conditions (McInerney and Wing, 2011). There were a number of smaller-magnitude hyperthermals in the early Eocene, but the PETM differs from these events with both higher magnitude and longer duration (Zachos et al., 2010; Bowen, 2013). However, there is still no consensus on the ultimate PETM cause, or whether several
mechanisms contributed to prolong the PETM duration (e.g. Zeebe et al., 2009; Bowen et al., 2015). Several $^{12}$C-enriched carbon sources may have contributed to the PETM CIE; the dissociation of methane clathrates (Dickens et al., 1995), a bolide impact activating terrestrial carbon reservoirs (Kent et al., 2003; Schaller et al., 2016), and volcanic and thermogenic degassing from the North Atlantic Igneous Province (NAIP; Fig. 1; Eldholm and Thomas, 1993; Svensen et al., 2004; Storey et al., 2007a).

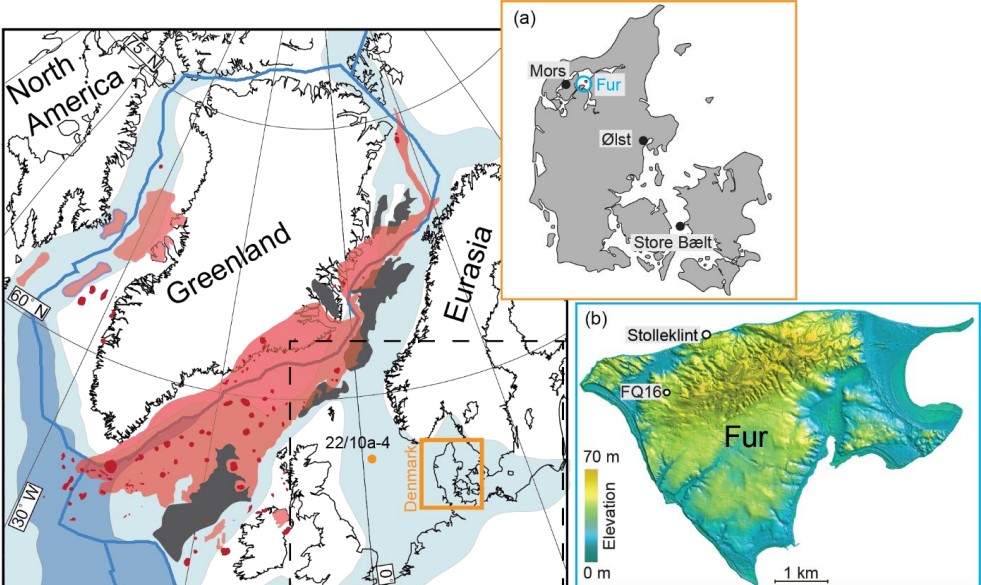

**Figure 1:** Plate tectonic reconstruction from 56 Ma with the known NAIP extent indicated (modified from Abdelmalak et al., 2016, Horni et al., 2017, and Jones et al., 2019). The orange dot notes the position of core 22/10a-4 described by Kender et al. (2012) and Kemp et al. (2016). Blue lines: plate boundaries. Black lines: present-day coastlines. Light and dark blue areas: shelf and deep marine areas, respectively. Light red areas: Known extent of subaerial and submarine extrusive volcanism from
the NAIP. Dark red: individual volcanic centres. Black areas: extent of known NAIP sill intrusions in sedimentary basins. The



total extent of intrusions beneath the extrusive volcanism is not known. The dashed square indicates the position of Figure 12. **a)** Map of Denmark with locations of Fur, Ølst and Store Bælt indicated **b)** Topographic map of the island of Fur (22 km²). The high topography in the north of the island is a partially overturned anticline of Fur Formation and upper Stolleklint Clay strata. Map courtesy of Egon Nørmark.

Marine uptake of increased atmospheric $CO_2$ altered the ocean chemistry, leading to deep ocean acidification and substantial deep-sea carbonate dissolution (Zachos et al., 2005; Babila et al., 2018). The temperature increase and ocean acidification were accompanied by transient ocean circulation changes, increased halocline stratification and a global reduction in bottom water oxygen (Kennet and Stott, 1991; Nunes and Norris, 2006; Kender et al., 2012; Pälike et al., 2014). This led to the extinction of 30–50 % of all benthic foraminifera species (Koch et al., 1992;

Thomas & Shackleton, 1996; Alegret et al., 2009; Nagy et al., 2013). Anoxic conditions were less extensive during the PETM than in previous ocean anoxic events (Jenkyns, 2010), and there were large regional variations in ocean oxicity (Pälike et al., 2014). Still, globally widespread ocean deoxygenation has been recognised (Pälike et al., 2014; Zhou et al., 2014; Yao et al., 2018), with particularly prevalent anoxic to euxinic conditions observed in semi-enclosed shelf areas such as in the Tethys Ocean (Egger et al., 2003; Khozyem et al., 2013), Peri Tethys

Basin (Gavrilov et al., 2003; Dickson et al., 2014), the North Sea (Schoon et al., 2015), and the Arctic Ocean (Stein et al., 2006; Harding et al., 2011).

The hydrological cycle also changed substantially during the PETM (e.g. Carmichael et al., 2017), with modelling studies suggesting an overall increase in extreme weather events (Carmichael et al., 2018). Proxy evidence indicates a more humid climate, particularly in higher latitudes and marginal marine areas such as Antarctica

(Robert and Kennett, 1994), the northeast US coast (Gibson et al., 2000; John et al., 2012), the Tethys (Bolle et al., 2000; Egger et al., 2003; Khozyem et al., 2013), North Atlantic (Bornemann et al., 2014), North Sea (Kender et al., 2012; Kemp et al., 2016), and the Arctic (Dypvik et al., 2011; Harding et al., 2011). In contrast, areas such as the Pyrenees (Schmitz and Pujalte, 2003) and the US interior (Kraus and Riggins, 2007) show evidence of arid climates. There seems to be considerable regional and temporal variation in the hydrological changes, with an

increased meridional transport of water vapour from low to high latitudes leading to an overall dry–dryer, wet–wetter climate response to the global warming (Carmichael et al., 2017).

The 4–5 °C PETM temperature increase (Dunkley Jones et al., 2013; Frieling et al., 2017) is comparable to that predicted in response to the current anthropogenic carbon emissions (e.g. Riahi et al., 2017). The PETM is therefore an important natural analogue for future greenhouse conditions, as the environmental and ecological response may

hold clues for the consequences of present day global warming (Zachos et al., 2010; Alley, 2016; Penman and Zachos, 2018; Svensen et al., 2019). Model predictions suggest that the current global warming will lead to an enhanced hydrological cycle, akin to that indicated by PETM proxy records (Held and Soden, 2006; Seager et al., 2010; Trenberth, 2011). The intensification of both droughts and extreme weather events are already occurring in parts of the world, with substantial consequences for human settlements (e.g. Riahi et al., 2017). Similarly, a

decrease in ocean oxygenation has been observed for the last 50 years, most likely resulting from the current global warming (Bograd et al., 2008; Stramma et al., 2012). The spread of marine anoxia is a well-known consequence of global warming, negatively affecting marine ecosystems as a whole (Stramma et al., 2008; Gilly et al., 2013). Understanding the timing and regional distribution of the environmental response to global warming in the past is therefore vital to meet the challenges of the future.

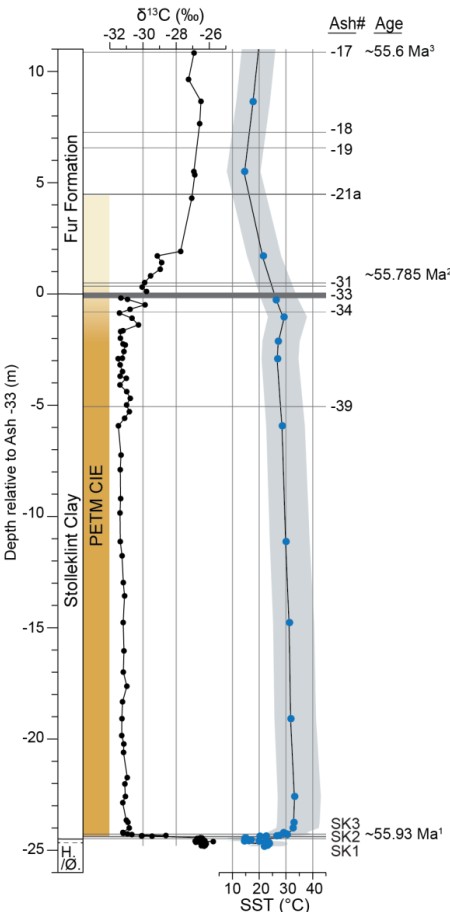


**Figure 2:** Data from the Stolleklint beach and a nearby quarry (Fig. 3), with lithological units indicated to the left. H./Ø.=Holmehus/Østerrende Formation. The depth scale is measured upwards and downwards from the main marker bed Ash -33. The yellow column indicates the interpreted PETM CIE duration, based on the $\delta^{13}C$ curve from Jones et al. (2019). The base of the column marks the CIE onset, and the gradually lighter colours toward the top marks the CIE recovery. Sea surface

temperature (SST) variations are given as BAYSPAR calibrated $TEX_{86}$ SSTs with 1σ error bars indicated by the light blue area (Stokke et al., 2020a). Ages are indicated based on [1]Westerhold et al., 2018; [2]Charles et al. (2011), assuming the Svalbard and Fur CIEs timing is coeval; [3]Age of Ash -17 from Storey et al. (2007a) recalibrated by Jones et al. (2019).

The Stolleklint section on the island of Fur in northwest Denmark offers an excellent opportunity to study the environmental response to temperature changes during the PETM in detail (Fig. 1). Denmark is placed in the

eastern part of the epicontinental North Sea, which during the latest Paleocene became a highly restricted basin due to NAIP thermal uplift (Knox et al., 2010). During the PETM the North Sea was characterised by bottom-water deoxygenation (Schoon et al., 2015), and a high sedimentary input, significant surface water freshening, and development of halocline stratification reflecting an intensified hydrological cycle (Zacke et al., 2009; Kender et al., 2012; Kemp et al., 2016). At Stolleklint, the PETM is recognized by a 4.5 ‰ CIE and the appearance of the

diagnostic dinoflagellate *Apectodinium augustum* at the base of the earliest Eocene Stolleklint Clay (Fig. 2;



Heilmann-Clausen, 1994; Schmitz et al., 2004; Schoon et al., 2013; Jones et al., 2019). The Stolleklint Clay –
which covers the PETM interval in Denmark – is a thermally immature and expanded clay-dominated section,
making this a unique and particularly well-suited section for detailed geochemical analyses. Located in a
downwind direction and within proximity to the NAIP, Denmark was also ideally placed to record the
contemporary volcanic activity from the NAIP (Fig. 1; Jones et al., 2019; Stokke et al., 2020b). This is evidenced
by the hundreds of NAIP tephra layers interbedded in the Danish and North Sea stratigraphy (Bøggild, 1918; Knox
and Morton, 1988; Larsen et al., 2003). The NAIP importance in the PETM initiation and termination is a topic of
much discussion (Svensen et al., 2004; Jolley and Widdowson, 2005; Storey et al., 2007a; Frieling et al., 2016;
Saunders, 2016; Gutjahr et al., 2017). To refine this relationship, better constraints on the relative timings of
volcanic activity and climatic and environmental changes are needed.

In our previous paper from Stolleklint, we presented a high resolution sea surface temperature (SST) reconstruction
based on the organic palaeothermometer TEX$_{86}$ (Stokke et al., 2020a). We found that SSTs increased by about 10
°C across the CIE onset, and then gradually decreased during the CIE body and recovery (Fig. 2; Stokke et al.,
2020a). Here, we combine mineralogical, sedimentological, and geochemical proxies to investigate the relationship
between changes in temperature and variations in both basin oxicity and sediment input; the latter typically inferred
to indicate changes in terrestrial erosion and runoff. Both increased weathering of siliciclastic rocks and enhanced
sequestration of organic carbon have been proposed as important negative feedback mechanisms, potentially
driving the PETM recovery (Bowen and Zachos, 2010; Ma et al., 2014; Penman, 2016; Dunkley-Jones et al., 2018).
Better constraints on the timing and global extent of increased silicate weathering and organic matter drawdown
are therefore vital for understanding the PETM termination.

## 2. Field area and stratigraphy

Stolleklint is located on the northern shore of the island of Fur in northwest Denmark (Fig. 1). In the Palaeogene,
Fur was part of the Norwegian–Danish Basin, a marginal basin in the eastern semi-enclosed epicontinental North
Sea Basin (Rasmussen et al., 2008; Knox et al., 2010). The Norwegian–Danish Basin forms a NW to SE striking
depression, bounded by the Fennoscandian Shield and the Sorgenfrei–Tornquist Zone to the NE and basement
blocks in the Ringkøbing–Fyn High to the SW (Schiøler et al., 2007). Salt diapirs of Zechsten salt creates additional
restricting structures within the Norwegian–Danish Basin (e.g. Petersen et al., 2008).

The base of the section at Stolleklint likely comprises the Holmehus Formation, which correspond to the Lista
Formation offshore in the North Sea (Figs. 2, 3, 4). This is a hemipelagic bioturbated fine-grained mudstone,
representing the culmination of a long period of transgression in the latest Paleocene Denmark (Heilmann-Clausen,
1995). In the latest Paleocene, a combination of thermal uplift around the NAIP (Knox, 1996) and tectonic uplift
along the Sorgenfrei–Tornquist Zone (Clausen et al., 2000) led to a relative sea-level fall and almost complete
isolation of the North Sea Basin (Knox et al., 2010). In Denmark, this resulted in either erosion of the latest
Paleocene strata, a hiatus in deposition, or deposition of the informal Østerrende clay unit above the Holmehus
Formation. However, the Østerrende clay unit has a very limited distribution, and it is uncertain how much is
present at Stolleklint despite its presence further south in Denmark (Schmitz et al., 2004; King, 2016). Schoon et
al. (2015) correlated the uppermost Paleocene stratigraphy at Fur with the Østerrende clay similar to that seen at
Store Bælt (Fig. 1). However, the Østerrende clay is absent in cores drilled at Mors ~20 km to the west and at Ølst



~80 km to the SE (Fig. 1A; Heilmann-Clausen, 1995), suggesting that a hiatus of uncertain duration followed the

Holmehus Formation at Stolleklint. Still, due to the uncertainty of this boundary, we will henceforth refer to the

lowermost unit as the Holmehus/Østerrende Formation.

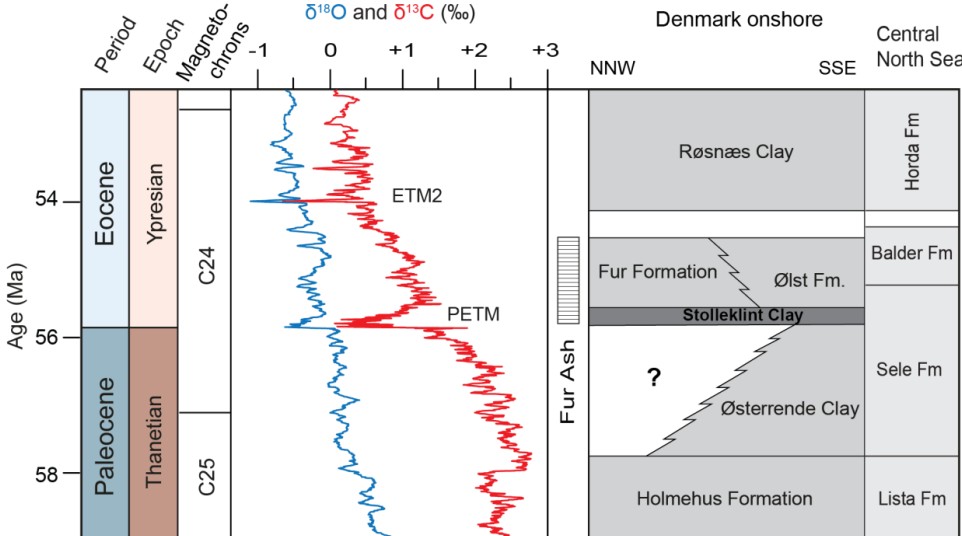

**Figure 3:** Composite figure of the late Paleocene and early Eocene interval, indicating both the Danish stratigraphy and the correlative offshore North Sea stratigraphy. Stratigraphy from King (2016) and Schiøler et al. (2007). The $\delta^{13}C$ and $\delta^{18}O$ curves

indicate the stratigraphic position of two periods of carbon perturbation; the Paleocene–Eocene Thermal Maximum (PETM) and the Eocene Thermal Maximum 2 (ETM2). Carbon and oxygen isotope data from Cramer et al. (2009) and Littler et al. (2014) and plotted on the GTS2012 timescale (Ogg, 2012).

The Paleocene–Eocene transition is seen as a lithological shift from the Holmehus/Østerrende Formation bioturbated clays, into the dark, laminated clays of the Stolleklint Clay (Fig. 4; Heilmann-Clausen et al., 1985;

Heilmann-Clausen, 1995; King, 2016). The lithological shift is accompanied by the almost complete absence of benthic fauna and preferential dissolution of remaining calcareous organisms within the Stolleklint Clay (Heilmann-Clausen, 1995; Mitlehner, 1996). The Stolleklint Clay is an informal unit, representing the lower Ølst Formation in northern Denmark and correlating with the offshore Sele Formation (Fig. 3; Heilmann-Clausen, 1995). A condensed, glauconite-rich silty horizon marks the Stolleklint Clay base (Heilmann-Clausen, 1995,

Schmitz et al., 2004; Schoon et al., 2015). This glauconite-rich silt contains mainly authigenic and biogenic grains and was likely deposited in an upper bathyal to outer neritic environment with low sedimentation rates (Nielsen et al., 1986; Schoon et al., 2015). A relative sea level rise is recorded in PETM sections in the Atlantic, Pacific, Tethyan, and Arctic Oceans (Sluijs et al., 2008; Harding et al., 2011; Pujalte et al., 2014; Sluijs et al., 2014). It was likely caused by thermal expansion of seawater due to global warming (Sluijs et al., 2008), and may pre-date the

PETM up to 20–200 kyr (John et al., 2012). Although this transgression was overprinted by regional tectonically forced regression in the latest Paleocene, the earliest Eocene Stolleklint Clay is deposited in an outer neritic environment (between 100–200 m; Knox et al., 2010; Schoon et al., 2015) during a gradual transgression (Heilmann-Clausen, 1995). The Stolleklint Clay is overlain by the ~60 m thick clay-rich Fur Formation diatomites (Figs. 2, 4), correlating to the offshore Sele and Balder Formations (Fig. 3). At Stolleklint, the PETM is defined



by a negative CIE of -4.5 ‰; characterized by a sharp onset above Ash SK2 at the base of the Stolleklint Clay, a thick stable body phase (~24 m), and a gradual recovery (~4.5 m) from about Ash -33 to around Ash -21a (Fig. 2, 4; Jones et al., 2019). The PETM at Stolleklint is associated with a substantially increased sedimentation rate from the condensed glauconitic base to a maximum sediment accumulation rate in the Stolleklint Clay of about 24 cm/kyr (Stokke et al., 2020a).

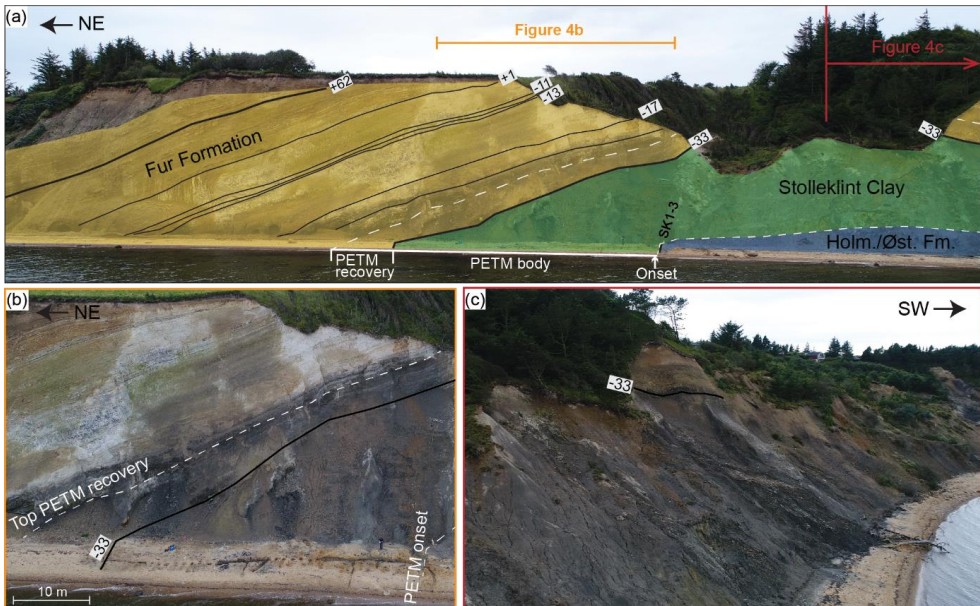

**Figure 4: a)** Picture of the coastal cliff at the Stolleklint beach with the main stratigraphic units shown as yellow (Fur Formation), green (Stolleklint Clay), and blue (Holmehus/Østerrende Formation). The black lines indicate certain key ash layers, and white dashed lines indicate the upper and lower bounds of the PETM CIE. **b)** Close up of the central part of the cliff face showing the colour difference between the dark PETM clays and the light post-PETM diatomites. Ash -33 and the PETM
CIE are indicated. Note the 43 m long ditch on the beach where most of the samples were collected. **c)** Picture of the continued cliff side towards the west, showing the extensive coastal erosion. Approximate location of Ash -33 is indicated.

More than 180 tephras with thicknesses up to 20 cm have been identified in the stratigraphy exposed at Fur, with the majority (~140) within the post-PETM Fur Formation (Fig. 3; Bøggild, 1918; Pedersen and Surlyk, 1983). Tephra is a general term for all air-borne volcanic fragmented material, but the grain sizes of all the Fur tephras
are <2 mm and therefore within the ash fraction. Heavily altered ashes are called bentonites, and while this applies to some of the lowermost ashes we will for simplicity use the term ash for all. The volcanic ashes are grouped in a negative and positive ash series based on variations on outcrop appearance and geochemistry (Bøggild, 1918), with additional ash layers SK1–4 identified later at the base of the Stolleklint Clay (Schmitz et al., 2004; Jones et al., 2019). The SK ashes and the negative series are a heterogeneous mix of ash compositions, whereas the positive
series are largely comprised of tholeiitic basalts (Morton and Evans, 1988; Larsen et al., 2003). All the ashes are believed to be sourced from NAIP explosive volcanism during the northeast Atlantic opening (Larsen et al., 2003; Storey et al., 2007a; Stokke et al., 2020b). These ash layers are found throughout the North Sea and North Atlantic



(Knox and Morton, 1988; Haaland et al., 2000) with some of the major layers traced all the way to Austria, suggesting that ash was occasionally distributed all over Northern Europe (Egger et al., 2000).

## 3. Materials and methods

### 3.1 Sampling

Samples were mostly collected from the Stolleklint beach (56°50'29''N, 8°59'33''E; Figs. 1B, 4), with some additional samples from a quarry near Fur Camping (Quarry FQ16 at 56°49'51''N, 8°58'45''E; Fig 1B). At Stolleklint, the Stolleklint Clay and the Fur Formation are exposed in the cliff side (Fig. 4). However, the base of the Stolleklint Clay and the Paleocene–Eocene transition were not exposed at the time of field work due to coastal erosion (Fig. 4C). We therefore excavated a 43 m long and 0.5 m deep trench along the beach (Fig. 4B). Recent glaciotectonic activity has resulted in a relatively steep bedding with internal small scale folding and thrusting (Pedersen, 2008), complicating stratigraphic thickness estimates. Jones et al. (2019) used trigonometry to estimate a local true thickness of $24.4 \pm 2$ m (24.2 m excluding ash layers) for the PETM onset and body at the Stolleklint beach; from the top of Ash SK2 to the base of Ash -33 within the excavated trench. This estimated true thickness is used as the depth scale for stratigraphic presentation (e.g. Fig. 2), measured as positive and negative depth relative to the base of the main marker bed Ash -33. Discrete samples were collected at 1 cm intervals from ~25 cm below to ~90 cm above Ash SK1 (in estimated true thickness), and then at 0.5 m intervals (0.2-0.3 m when converted to the estimated true thickness) up to Ash -33. Samples above ash -33 were collected from the cliff face at Stolleklint at ~10–20 cm intervals. Additional samples from -5.6 to +1.9 m relative to the base of Ash -33 were included from the quarry FQ16, sampled at ~30 cm intervals. All samples were oven dried at ≤50 °C and powdered in an agate hand mortar or an agate disc mill before further analysis.

The sediments unconsolidated character enabled the collection of 4 box-cores up section. The box-cores were collected in 50 cm long, and 5 cm wide and deep aluminium boxes. These were pushed into the sediments before surrounding material was removed and the box cut away with its content intact using a steel wire. Box-cores were collected in order to get complete recovery of selected intervals for XRF core scanning (Fig. 5). Two box-cores were collected across the PETM onset (-24.90 to -24.40 m and -24.63 to -24.20 m stratigraphic depth intervals), and two from the PETM body with one from the lower laminated part (-14.47 to -14.17 m) and one from the non-laminated upper part (-10.81 to -10.48 m).

### 3.2 XRD bulk and clay mineralogy

Bulk rock mineralogy was conducted on 8 samples from -24.81 to +5.35 m depth, while 13 samples were analysed for clay minerals. The mineralogy of both bulk rock and clay fraction (< 2 µm) of Fur sediment samples were determined by X-ray diffraction (XRD) analyses on a Bruker D8 ADVANCE diffractometer with a Lynxeye 1-dimensional position-sensitive detector (PSD) and CuKα radiation ($\lambda$ = 0.154 nm; 40mA and 40kV) at the Department of Geosciences, University of Oslo. The bulk-rock fraction was wet milled in a McCrone Micronizing Mill, prepared as randomly oriented samples, and analysed with a step size of 0.01° from 2° to 65° (2θ) at a count time of 0.3 s (2θ). The software DIFFRAC-EVA (v. 2.0) was used for phase determination, and phase quantities were determined by Rietveld refinement (Rietveld, 1969) using PROFEX (v. 3.13.0; Doebelin and Kleeberg, 2015).



The clay fraction (< 2 µm) was separated from the crushed whole-rock sample (before wet milling) by gravity
       settling, and then prepared as oriented aggregate mounts using the Millipore filter transfer method (Moore and
       Reynolds, 1997). As the dried samples had to be powdered prior to separation, they contain some minor
       contribution from the coarser fraction. XRD clay data were recorded with a step size of 0.01° from 2° to 65° (2θ)
       at a count time of 0.3s (2θ) in air-dried samples, and a step size of 0.01° from 2° to 34° (2θ) at a count time of 0.3s
(2θ) on treated samples. Three rounds of treatments were applied: 24h of ethylene glycol saturation, 1h heating at
       350 °C, and 1h heating at 550 °C. The software NewMod II (Reynolds and Reynolds, 2012) was used for semi-
       quantification of the XRD-patterns of inter-stratified clay minerals.

### 3.3 XRF core scanning

       Non-destructive geochemical measurements and radiographic images were obtained from the box-cores with an
ITRAX X-ray fluorescence (XRF) Core Scanner (Croudace et al., 2006) from Cox Analytical Systems at the
       EARTHLAB facilities, Department of Earth Science, University of Bergen. The core scanner was fitted with a
       molybdenum X-ray tube run with power settings at 30 kV and 30 mA. The box-cores were scanned with 10 s
       exposure time at 0.5 mm sampling intervals.

### 3.4 Rock-Eval pyrolysis

The bulk organic-carbon characteristics including the hydrogen index (HI), oxygen index (OI), and $T_{MAX}$ were
       determined by analysing ca 50 mg powder aliquots with a Rock-Eval 6 (Vinci Technologies SA, Nanterre, France;
       Behar et al., 2001) at the University of Oxford. A total of 39 samples were analysed between -24.81 and +0.01 m
       depth. The HI corresponds to the quantity of hydrocarbons per gram TOC, and reflects the relative distribution of
       terrestrially and marine derived organic matter. A HI <100 indicates a dominantly terrigenous source, while a HI
>>100 indicates the presence of significant amount of aquatic algae (marine and/or freshwater) and/or microbial
       biomass (e.g. Stein et al., 2006).

### 3.5 ICP-MS and Element Analyser

       Analyses were conducted on 24 samples between -24.82 to +5.50 m depth. Dried and crushed marine sediment
       samples were digested in hydrochloric, hydrofluoric, and nitric acids. Major and trace element analyses of digested
bulk sediment samples were performed on a PerkinElmer NexION 350D ICP-MS. Total sulfur concentrations were
       analysed on a Coulomat 702 coulometric analyser. Sample digestion and analyses were all conducted at the
       Department of Earth Sciences, University of Oxford. The method detection limit is indicated in the supplementary
       material (Table 4, Supplement 1). $P_2O_5$, Ba, Cu, Ni, and V have all been normalized to $Al_2O_3$ to account for the
       potential detrital influx, as $Al_2O_3$ indicates the aluminosilicate fraction of the sediments (Tribovillard et al., 2006).

ICP-MS analyses of major elements were used to calculate the chemical index of alteration (CIA; Nesbitt and
       Young, 1982). The CIA is a measure of weathering intensity based on the relative distribution of mobile cations
       relative to aluminium oxide, and indicates the extent of conversion of feldspars (which dominate the upper crust)
       to clays such as kaolinite (Nesbitt and Young, 1982). While the CIA may directly represent the rate of weathering
       when measured *in situ*, when measured in marine sediments it becomes more complex as it also reflects the type
of sediment and the transport sorting processes (Eq. 1; Nesbitt and Young, 1982). The CIA is expressed as:

$$CIA = \left( \frac{Al_2O_3}{Al_2O_3 + CaO + K_2O + Na_2O} \right) * 100 \qquad (1)$$





Where $Al_2O_5$, CaO, $K_2O$, and $Na_2O$ are given as whole-rock molecular proportions.

## 4. Results

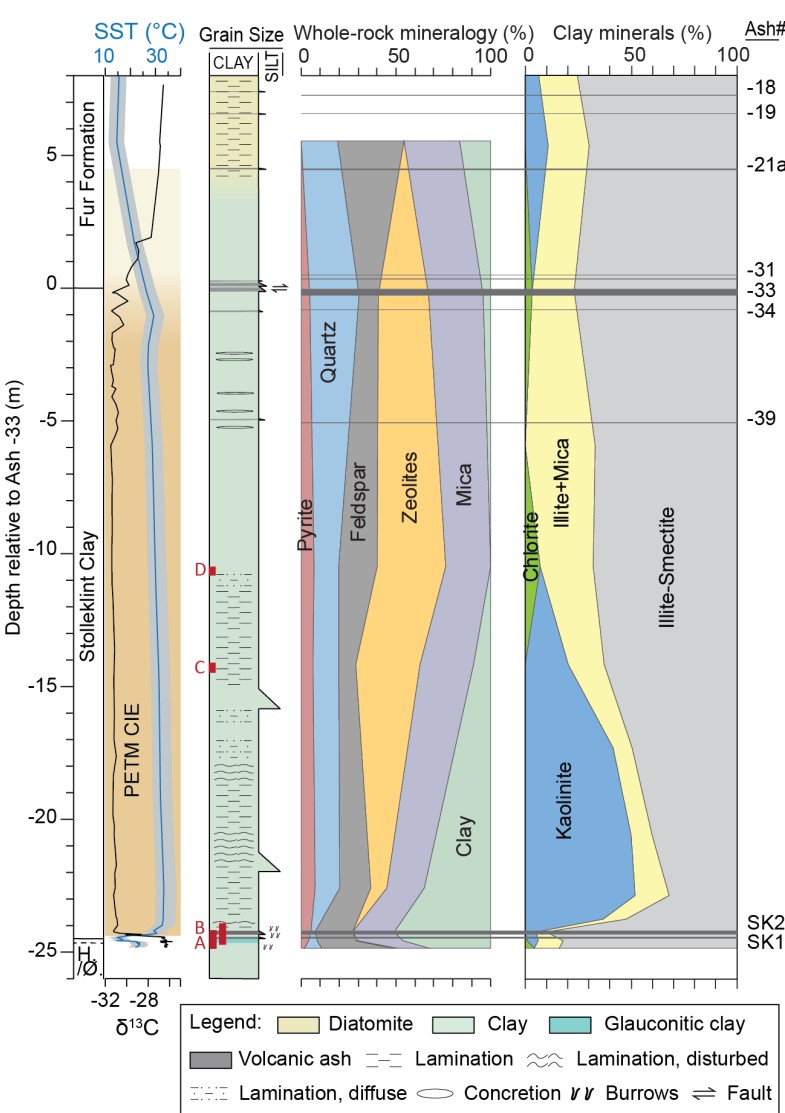

**Figure 5:** Sedimentological log from Stolleklint with legend below. Red squares labelled A–D indicate the XRF box-core scans position in Figure 6. Bulk-rock and clay mineralogy is presented as percentage. $\delta^{13}C$ data from Jones et al. (2019), $TEX_{86}$ data from Stokke et al. (2020a).

### 4.1 Sedimentology

The base of the beach section comprises the Holmehus/Østerrende Formation (Figs. 4, 5), which is composed of dark, blueish clay with pervasive bioturbation. It is overlain by a greenish silty layer indicative of glauconite



(marked with G in Fig. 6), with up to coarse sand-sized aggregates of glauconite scattered within the clay. The silts lower boundary is unclear, but it appears conformable and possibly gradational. The 5 cm thick ash layer SK1 is deposited above the glauconitic silt, with a sharp undulating lower boundary (Fig. 6). About 4.5 cm of structureless, grey clay conformably overlies SK1, and is followed by the ~8 cm thick ash layer SK2 (Fig. 6). Both ashes are

light grey and seems to be heavily altered. They are upward fining from medium sand to clay-sized particles. Both ashes are relatively reworked and become gradually more clay-rich toward the top, with the highest bioturbation intensity at the top of Ash SK2 (Fig. 6). About 2 cm of strongly bioturbated and ash-rich clay overlying the ash is abruptly ended by the initiation of dark laminations (Fig. 6 section B). The exact level of the Stolleklint Clay base is uncertain as the boundary is blurred by ashes SK1 and SK2, but the start of the laminations is included in the

Stolleklint Clay, placing the boundary no higher than here (Fig. 6 section B). Laminated dark clay continues for ~10 cm before deposition of two ash layers SK3 and SK4, ~1 cm and ~0.4 cm thick respectively (Fig. 6 section B). They are separated by 2 cm of clay with slightly undulating lamination. Above the ash, laminated clay continues about half-way up the beach (Fig. 5), with increasingly folded and disturbed layering (Fig. 7 section C).

The PETM body is dominated by hemipelagic clay. Above the lower laminated part, it appears to have an upper

part (from about -10 m depth) comprising very dark grey clay with no visible laminations in field exposures (Fig. 5). However, the XRF radiographic image reveals that there are intermittent diffuse laminations and patchy structures/colour differences within the clay (Fig. 8 section D). The cause of these colour patches is uncertain, but could be a result of depositional variations and/or post-depositional deformation. Between about -6 and -2 m depth there are some highly pyritized concretions, or likely broken up concreted layers (Fig. 5). Ash layers reappear from

about -5 m depth with deposition of the thin (~2 cm), black Ash -39 (Fig. 5). Ashes -34, -33, -32, and -31 are deposited relatively closely spaced between -0.85 to +0.05 m depth, with thicknesses of 2, 20, 2 and 3 cm respectively. The thickest layer Ash -33 is repeated at the Stolleklint Beach, due to a small glaciotectonic thrust fault (Fig. 5). The boundary between the Stolleklint Clay and the Fur Formation is formally placed at Ash -33, although there is no sharp lithological boundary (Figs. 4, 5). Dark clays continue upward with a gradually

increasing diatomite content. Laminations re-appear at about +6 m depth, as the lithology become dominated by clay-rich diatomite (Fig. 5).

### 4.2 Mineralogy

The bulk mineralogy comprises six main phases: zeolites, mica, clay (including smectite, chlorite, kaolin minerals, and glauconite), feldspars, quartz, and pyrite (Fig. 5). Phyllosilicates dominate the bulk mineralogy in the lower

laminated part of the stratigraphy. While the mica fraction remains relatively stable throughout, the clay fraction reaches its maximum of 50.6 % at ~13 cm above Ash SK2 and the CIE onset (-24.24 m depth), before decreasing substantially upward from about -22 m depth (Fig. 5). Zeolites (of the heulandite–clinoptilolite type) dominate the CIE body, comprising up to 36 % of the bulk mineralogy at -10.48 m depth (Fig. 5). The fraction of feldspars is largest within the Holmehus/Østerende Formation (37 % at -24.81 m depth) and during the CIE recovery (35 % at

+5.35 m depth), while quartz increases in the upper part of the stratigraphy up to 26 % at -0.28 m depth (Fig. 5). Pyrite makes up the smallest fraction of the bulk mineralogy (Fig. 5). It increases from 1.9 % in the lower Holmehus/Østerrende Formation (-24.81 m depth) to 5.3 % ~13 cm above the CIE onset (-24.24 m depth). The highest fraction of pyrite (6.1–7.5 %) is reached during the CIE body, before values decrease during the PETM recovery to a minimum of 0.11 % at +5.35 m depth.



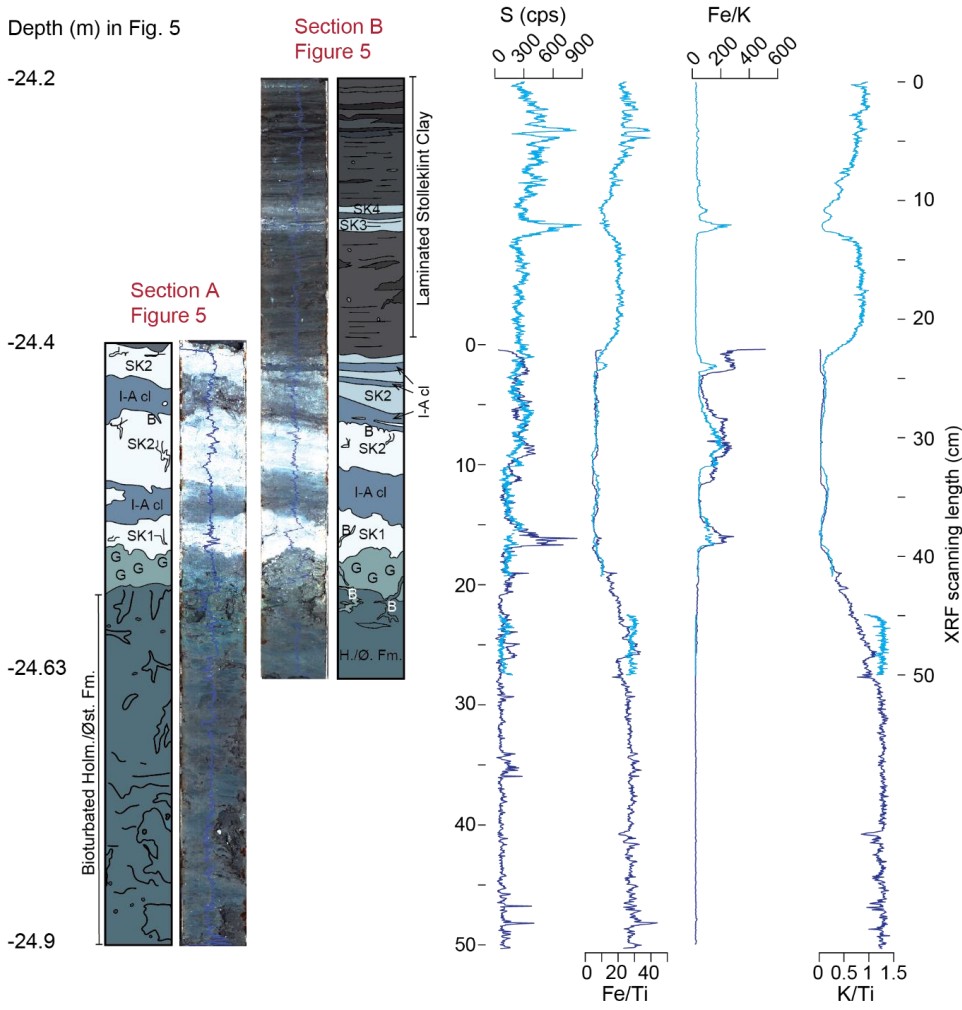

**Figure 6:** XRF element core scans and radiographic images of two box-cores crossing the PETM onset at the Stolleklint beach. Interpretive logs next to the images indicate the lithological changes. G=Glauconite; B=Burrow; I-A cl=Inter-ash clay. The corrected stratigraphic depth relative to Ash -33 of each section is indicated to the left. XRF scanning length seen to the right indicate actual box-core length. XRF data given as counts per second (cps) or as dimensionless ratios.

Clay fraction XRD analyses identified four major clay mineral phases: kaolinite, chlorite, mixed-layer illite–smectite with only minor illite layers indicating almost pure smectite, and illite + fine-grained mica (Fig. 5). Illite–smectite is the dominating clay mineral within the studied section. It comprises 84–90 % of the total clay from the base and up to 13 cm above the CIE onset (-24.24 m depth), before decreasing in the lower PETM body to a minimum of 32 % about 1.5 m above the CIE onset (-22.86 m depth). The illite–smectite content increases throughout the upper CIE body and recovery with values between 50–77 % (Fig. 5). Illite + fine-grained mica comprises a smaller part of the total clay fraction, with 10 % during the CIE onset, and a maximum of 33 % at -5.93 m depth (Fig. 5). Kaolinite increases substantially from 5 % about 13 cm above the CIE onset (-24.24 m depth) to 37 % at 62 cm above the CIE onset (-23.75 m depth). Kaolinite dominates the clay fraction in the lower





laminated PETM body with a maximum of 52 % at -20.60 m depth, before disappearing in the upper PETM body

and re-emerging with 11 % during the recovery phase (+5.35 m depth; Fig. 5). Chlorite only appear in 4 of 13

samples and makes up the smallest part of the clay fraction, with a maximum of 7 % at -10.48 m depth (Fig. 5).

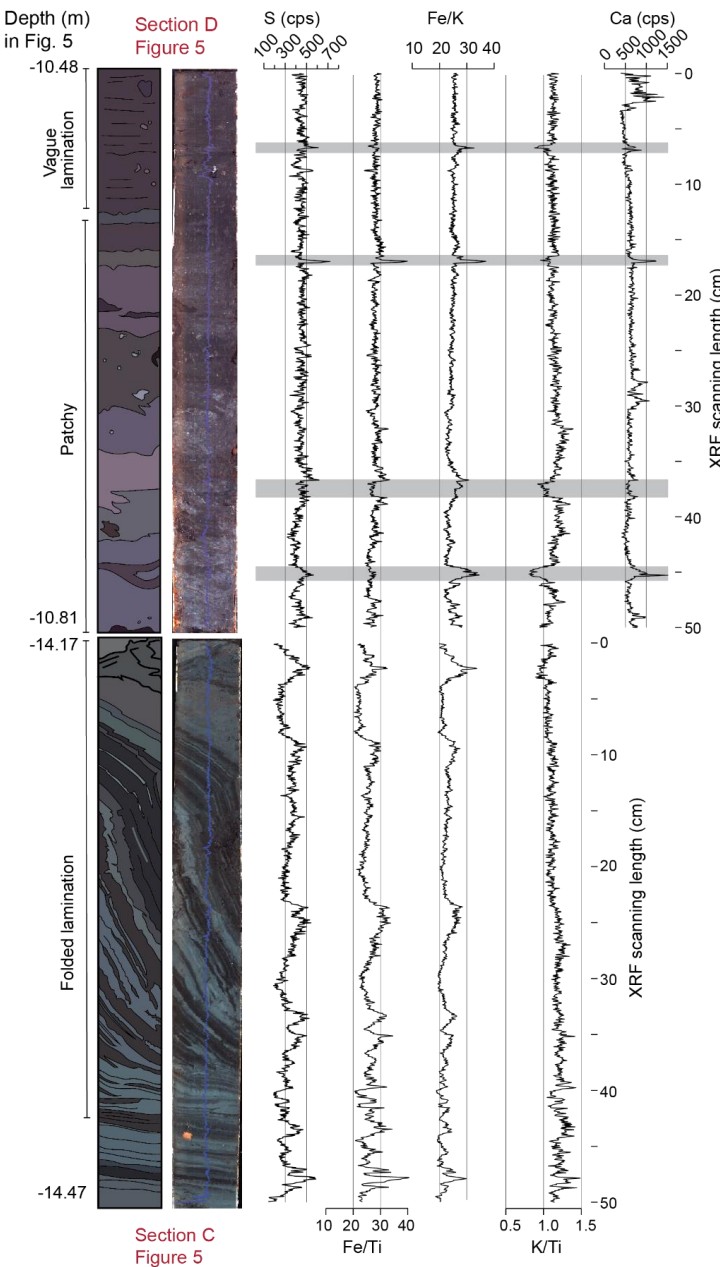

**Figure 7** XRF element core scans and radiographic images of two box-cores within the PETM body from the Stolleklint beach.
Interpretive logs next to the images indicate the lithological changes. The corrected stratigraphic depth relative to Ash -33 for





each section is indicated to the left, while the XRF scanning length to the right indicate actual box-core length. Grey bands in section D indicate potentially tephra-rich horizons. XRF data given as counts per second (cps) or as dimensionless ratios.

**4.3 XRF core scanning**

Two box-cores cross the PETM onset, covering the transition from Holmehus/Østerrende Formation into Stolleklint Clay, and the ash layers SK1–SK4 (Fig. 6). Sulfur counts show a slight overall increase from below to
above the ashes, suggesting gradually more suboxic conditions above the ashes. Low K/Ti and Fe/Ti ratios suggest that the ashes are Ti-rich basalts. The gradual decrease in both K/Ti and Fe/Ti below Ash SK1 may subsequently suggests a potential gradual increase in volcanic-derived material before the first ash layers in the Danish Basin appears (Fig. 6). Peaks in S counts indicate particularly S-rich parts of the ashes, although when correlating with Fe/Ti peaks it is more likely due to suboxic formation. Above Ash SK3, peaks of S, Fe/Ti, and Fe/K (although the
latter signal is swamped by the iron-rich ashes in Fig. 6 section B) correlate with the dark lamination, suggesting at least periods of decreased oxicity.

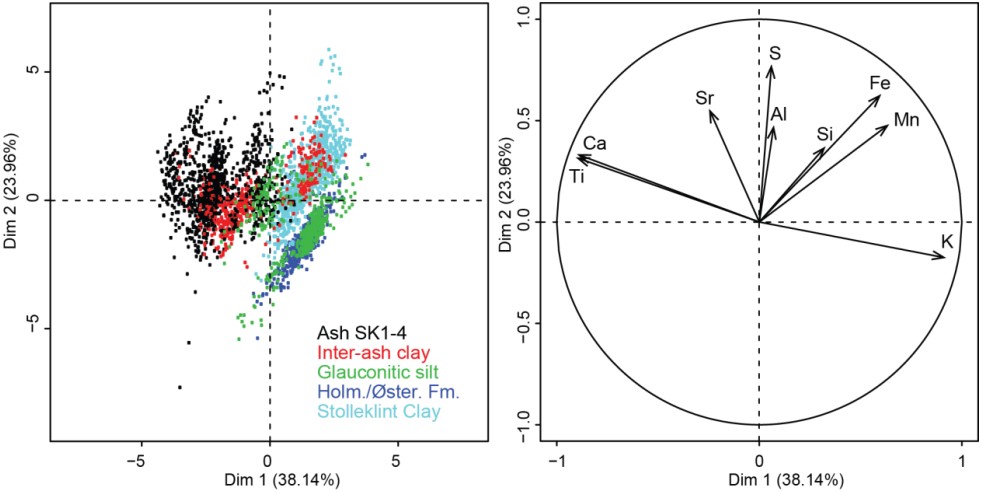

**Figure 8**: Biplot (left) and correlation circle (right) with Dimensions 1 and 2 (denoted as Dim 1 and Dim 2) of the principal component analysis applied to XRF core scanning data from box cores 1 and 2 across the PETM onset. Dimensions 1 and 2
represent 62 % of the total variability. The inter-ash clay in the Biplot refers to the clay between ashes SK1 and SK2 as indicated in Figure 7.

Principal component analysis reveal a distinct difference in chemistry between the Holmehus/Østerrende Formation and the Stolleklint Clay (Fig. 8). It also indicates that both the clay between ashes SK1 and SK2 and parts of the glauconitic silt likely include a large ash component. While the glauconitic silt is chemically closer to
the underlying Holmehus/Østerrende Formation than the Stolleklint Clay, the less ash-rich inter-ash clay appears to have a composition closest to the Stolleklint Clay. This suggests that this is indeed a part of the Stolleklint Clay base, and we therefore propose that Ash SK1 marks the lower Stolleklint Clay boundary. The correlation circle indicate that differences in Ca and Ti on one hand and K on the other is the main controlling factors (Fig. 8).





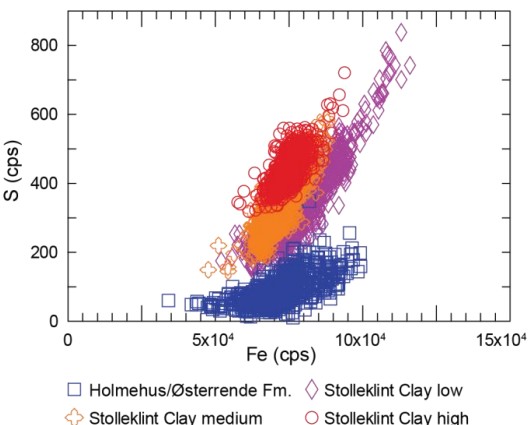

**Figure 9:** Biplot of Fe and S. Data from XRF element core scans of all four box-cores. CPS = Counts Per Second.

The box core in Figure 7 section C, covering the lower Stolleklint Clay, shows that the sediments are strongly laminated and slightly folded. Elevated S concentrations and high Fe/Ti and Fe/K ratios indicate anoxic conditions. Correlations between these peaks in anoxia and dark laminations (Fig. 7 section C) suggest that there were regular variations in bottom-water oxicity. The K/Ti ratio remains relatively stable, suggesting no dramatic lithological changes. Figure 7 section D shows the non-laminated upper Stolleklint Clay, which displays relatively minor elemental variations. However, drops in the K/Ti ratio could indicate areas of increased volcanically-derived material, potentially as cryptotephras (Fig. 7 section D); defined as volcanic tephra deposits not visible to the naked eye. The presence of cryptotephra layers is particularly likely when low K/Ti ratios correlate with increases in Fe/Ti and Ca, and to some extent S. Relative changes in Fe/Ti – and to some extent Ca – depend strongly on the source of the volcanic material. The biplot of S and Fe (Fig. 9) shows that the variability of S measurements decreases upward, and that there is an overall increase in S upward from the pre-PETM Holmehus/Østerrende Formation and throughout the Stolleklint Clay.

### 4.4 Major and trace elements of single samples

#### 4.4.1 Detrital input

The CIA in the pre-PETM sediments is generally at around 75, but has one peak of 85 just before the pre-PETM cooling event (-24.64 m depth; Fig. 10) indicating a relative rise in the influx of terrestrially weathered material. Following the onset, the CIA increases to a maximum of about 83 at -20.60 m depth, before returning to pre-PETM values in the upper PETM body (Fig. 11). The recovery phase shows increasing CIA values again towards Ash -19, with 86 at +5.50 m depth.

The sediments thermal immaturity was verified by the low RockEval $T_{max}$ values of <422 °C (Table 3, Supplement 1). The HI peaks up to 150 mgHC/gC in the glauconitic silt between -24.61 to -24.59 m depth, but is otherwise <100 mgHC/gC pre-PETM (Fig. 10). The HI increases >100 mgHC/gC about 13 cm above the CIE onset at -24.05 m depth. A second major increase in HI occurs above -14 m depth, after which values remain high and reaches maximum values of 303 mgHC/gC at -0.78 m depth (Fig. 11).



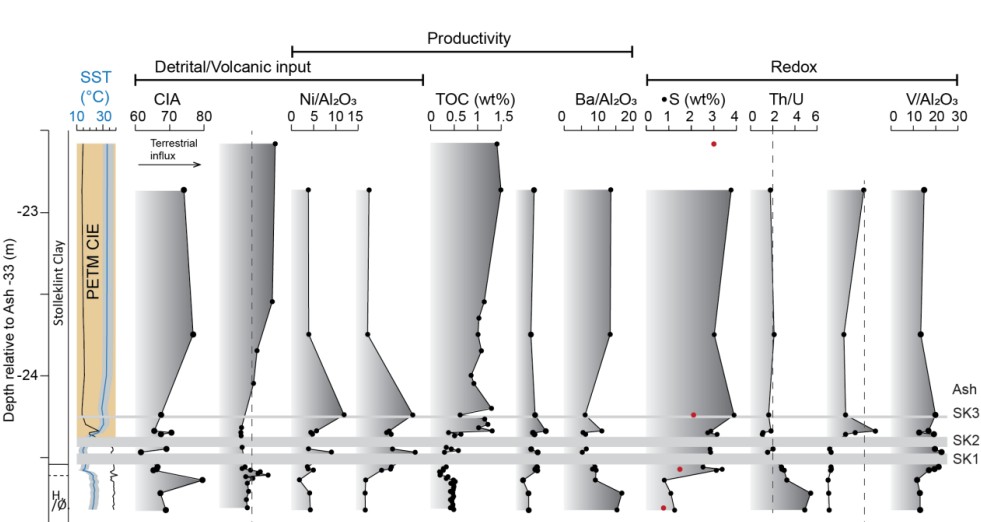

**Figure 10:** The lowermost 2.5 m of the Stolleklint beach section showing a close-up of the CIE onset. Graphs show: $TEX_{86}$ data from Stokke et al. (2020a); $\delta^{13}C$ and Total Organic Carbon (TOC) data from Jones et al. (2019); Chemical Index of Alteration (CIA); Hydrogen Index (HI) where vertical line divide between dominantly terrigenous (<100) and marine (>100) organic material; productivity proxies Ni, Cu, $P_2O_5$, and Ba normalised to $Al_2O_3$; fraction of pyrite from bulk XRD analyses; S concentrations in wt%; Th/U ratio where vertical line divide anoxic (<2) form oxic (2–7) sediments; molybdenum concentrations in ppm where vertical line divide between anoxic (<30 ppm) and euxinic (>30 ppm); V normalised to $Al_2O_3$.

### 4.4.2 Productivity

The TOC data shows a pronounced increase from ~0.45 to ~1.3 wt% TOC about 2 cm above the PETM CIE onset (Fig. 10). TOC concentrations remain relatively stable for the lower CIE body, before increasing again in the upper CIE body (from about -13 m depth) up to a maximum of 3.9 wt% at -0.78 m depth (Fig. 11). At the start of the CIE recovery, TOC drops down again to around 1 wt%.

Before the CIE onset $Ba/Al_2O_3$ decreases from the base of the glauconitic silt, while $Cu/Al_2O_3$ increases, and $Ni/Al_2O_3$ and $P_2O_5/Al_2O_3$ remain relatively stable (Fig. 10). Both Cu and Ni are elements typically associated with volcanic ash, and both remain relatively high within the ash-rich interval. The main increase in productivity proxies occurs within the uppermost CIE body between about -8.56 m depth and Ash -33 (Fig. 11). This trend is most notable in $Ba/Al_2O_3$ and $P_2O_5/Al_2O_3$, reaching maximum values of 49.8 and 0.017 respectively at -4.48 m depth. $Cu/Al_2O_3$ shows a similar although less distinct trend with the main increase in the upper CIE body, while $Ni/Al_2O_3$ decreases much earlier at about -5 m depth. During the recovery $Ba/Al_2O_3$ and $P_2O_5/Al_2O_3$ decreases, while $Cu/Al_2O_3$ and $Ni/Al_2O_3$ remain relatively stable (Fig. 11).



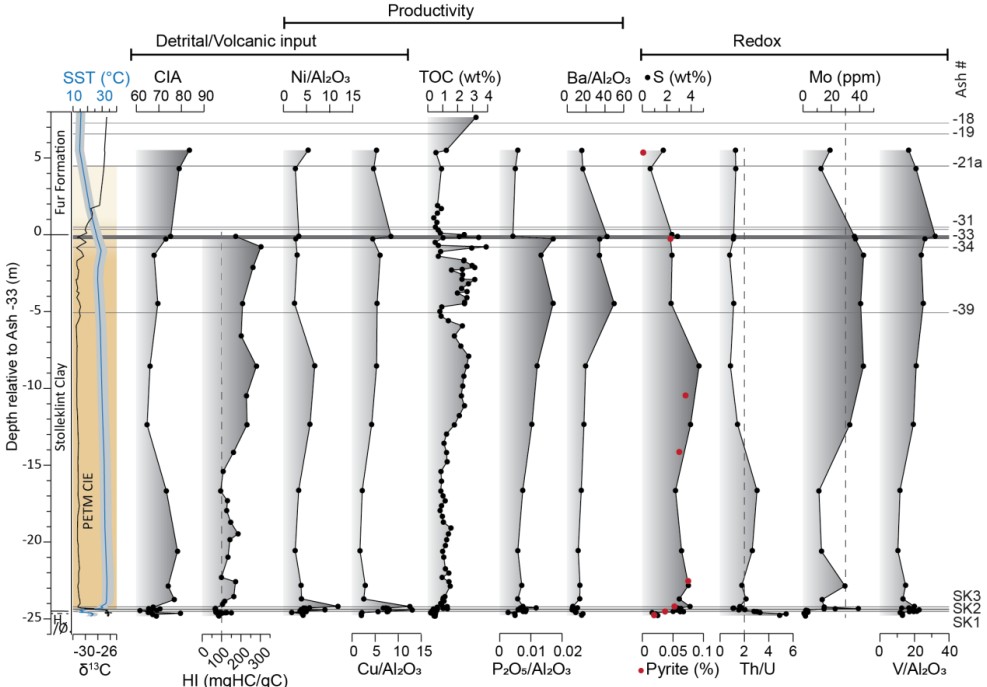

**Figure 11:** The section covering the whole Stolleklint PETM, with some additional samples from the nearby quarry FQ16 (Figure 2) between -5.6 and 2 m depth. See Figure 10 for details on each graph.

### 4.4.3 Redox

Both S and pyrite concentrations start to rise before Ash SK1 and the CIE onset, with S increasing from about 1 wt% in the Holmehus/Østerrende Formation to about 3 wt% in the glauconitic silt (Fig. 10). Sulfur concentrations remain high throughout the CIE body, with maximum values of 4.6 wt% reached at -8.6 m depth (Fig. 11). Th/U decreases below Ash SK1 and reaches values below 2 before the CIE onset (Fig. 10). In the lower CIE body between -20.60 to -16.68 m depth Th/U increases up to 3, coinciding with slightly lowered S concentrations (Fig. 11). Th/U values decreases again and persist below 2 for the remaining CIE body (Fig. 11), as U concentrations are enriched up to 11.4 ppm (Table 4, Supplement 1), well above average crustal values of 2.8 ppm (Taylor and McLennan, 1995).

Molybdenum concentrations vary between 0.7–3.0 ppm below the CIE onset, before increasing to ~15 ppm about 2 cm above the CIE onset (Fig. 10). Molybdenum concentrations continue between 11–43 ppm indicating anoxic conditions prevail throughout the CIE body and recovery, with euxinic conditions (>30 ppm Mo) indicated in the upper CIE body between -12.4 to -0.1 m depth (Fig. 11).

V/Al$_2$O$_3$ increases in the glauconitic silt and remain relatively high in the ash interval (Fig. 10). Vanadium enrichments can also be associated with ash deposition, which might affect V concentrations in this particularly ash-rich interval. V/Al$_2$O$_3$ values increase gradually throughout the CIE body, reaching maximum just below Ash -33 at -0.11 m depth, and decreases again in the recovery (Fig. 11).





### 5. Discussion

#### 5.1 Changing sediment input – tectonic and climatic influence

##### 5.1.1 Illite–Smectite – importance and origin

Smectite is the dominant clay mineral within the pre-PETM and most of the earliest Eocene strata at both Fur (Fig. 5) and generally in the North Sea (Nielsen et al., 2015). Clay mineral assemblages have been used as indicators of palaeoclimate, most commonly using kaolinite as a proxy for humid tropical climates and smectite for warm climates with seasonal humidity and longer dry spells (e.g. Thiry, 2000). However, soil formation is a slow processes, and the subsequent long duration between formation and deposition in a marine basin suggests that clay mineralogy is an unreliable palaeoclimate proxy at resolutions shorter than 1 Myr (Thiry, 2000). Changes in the clay mineral assemblage in the marine sediments may therefore instead indicate changes in source area and intensity of sediment transport, and reflect the climatic conditions at the time of continental soil formation rather than at the time of deposition (Thiry, 2000; Nielsen et al., 2015). Smectite is a common weathering product of mafic volcanic material (Stefánsson and Gíslason, 2001), and previous studies have suggested that smectites in the Danish stratigraphic record are of predominantly volcanic origin (Nielsen and Heilmann-Clausen, 1988; Pedersen et al., 2004). Although smectite may precipitate *in situ* from hydrothermal fluids, this has largely been discounted in the North Sea due to the wide geographic extent of smectite and the overall lack of indices of hydrothermal influence (Huggett and Knox, 2006; Kemp et al., 2016). *In situ* post-depositional alteration of volcanic ash also probably contributed only minor amounts of smectite, as the ashes are mostly relatively well-preserved (Nilsen et al., 2015). While clay minerals make up a small fraction of the bulk mineralogy in the upper PETM body (4–8 %), zeolites comprise up to 36 % (Fig. 5). Zeolites are another typical weathering product of volcanic materials, supporting a volcanic provenance (Stefánsson and Gíslason, 2001; Nielsen et al., 2015).

The volcanic source is likely to be the NAIP. Major flood basalts erupted in East Greenland, the Faroe Islands, and the British Isles across the Paleocene–Eocene boundary, producing large uplifted areas several km high of easily eroded material (Larsen and Tegner, 2006; Storey et al., 2007b; Wilkinson et al., 2017). This is reflected in Os isotopes and CIA records in the Arctic Ocean, which record an influx of weathered volcanic material both prior to and during the PETM (Wieczorek et al., 2013; Dickson et al., 2015). Moreover, the first phase of ash deposition was identified within Late Paleocene strata in the North Sea, well before the PETM onset (Knox and Morton, 1988; Haaland et al., 2000). Erosion and redeposition of altered tephra likely constituted a highly important source for the volcanic material in the North Sea (Pedersen et al., 2004; Nielsen et al., 2015; Kemp et al., 2016). Smectite is found in abundance throughout the North Sea stratigraphic record, and decreases as ash deposition ceases upward in the Eocene (Nielsen et al., 2015; Kemp et al., 2016). It seems likely that the dominance of smectite and abundance of zeolites reflect this extensive extrusive volcanism around the NAIP (Nielsen et al., 2015; Kemp et al., 2016).

##### 5.1.2 Kaolinite and changes in weathering across the PETM onset

An important deviation from the smectite dominance is the substantial influx of kaolinite in the lowermost 10 m of the PETM CIE (Fig. 5). The abundance of kaolinite at Fur rises shortly after the CIE onset and again in the CIE recovery, in both instances concordant with an increase in the CIA and the overall clay fraction (Figs. 5, 11). An increase in kaolinite content during the PETM is observed globally (Robert and Kennett, 1994; Dypvik et al., 2011;



John et al., 2012; Khozyem et al., 2013; Bornemann et al., 2014; Kemp et al., 2016). While it was initially attributed to a warmer and more humid climate (e.g. Bolle et al., 2000), it is now generally acknowledged that the formation

of kaolinite requires too much time (1–2 Myr; Thiry, 2000) to be a direct result of climate on such short timescales (Carmichael et al., 2017). The increase in kaolinite is therefore most likely due to an intensified hydrological cycle, leading to enhanced erosion and sediment transport of older deeply weathered bedrock and soils (Schmitz and Pujalte, 2003; John et al., 2012; Bornemann et al., 2014).

Although the kaolinite pulse appears to be global, the timing and magnitude varies a great deal even within the

North Sea (Kender et al., 2012; Kemp et al., 2016). In the western North Sea, the kaolinite content increases earlier before and during the CIE onset and again similarly during the CIE recovery, but is relatively low in the CIE body (Kender et al., 2012; Kemp et al., 2016). However, at Fur a kaolinite content increase is not observed until after the CIE onset (Fig. 5), and southward in the Bay of Biscay in the North Atlantic the kaolinite content does not rise at all until the PETM recovery (Bornemann et al., 2014). It would be expected that changes in the climate and the

hydrological cycle would be broadly similar within such a narrow region. It is therefore reasonable to assume that the timing and extent of kaolinite deposition depends just as much on the availability and proximity to potential source areas as the climatic conditions.

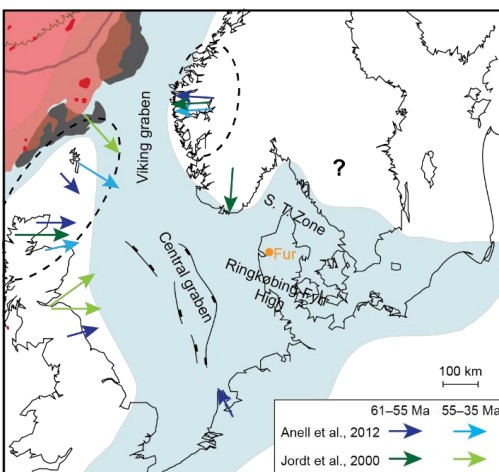

**Figure 12:** Close up of Figure 1 with modifications based on Schiøler et al. (2007). Dashed circles indicate main sedimentary

source areas for the North Sea during the Paleocene and Eocene from Anell et al. (2012). Arrows indicate the main sediment transport directions based on Jordt et al. (2000) and Anell et al. (2012).

During the Paleocene and Early Eocene, the major sediment transport direction in the North Sea were from the west and northwest due to the thermal uplift of source areas around the NAIP such as the Faroe–Shetland platform (Fig. 12; Knox, 1996; Anell et al., 2012). Kender et al. (2012) subsequently suggested that the initiation of the

kaolinite pulse before the CIE in the central North Sea was due to the thermal uplift and short-lived regression in the latest Paleocene. At Fur, there is no pre-PETM pulse of kaolinite and the HI index is >100 within the glauconitic silt (Fig. 10), indicating an increase in aquatic organic matter deposition. However, it seems there is also an augmented CIA just below the glauconitic silt (~28 cm below the CIE onset at -24.64 m depth; Fig. 10), which suggests some increase in the fluxes of terrestrially-derived material could have occurred also prior to the CIE



onset at Fur. Stokke et al. (2020a) similarly observed a peak in terrestrially derived long chain *n*-alkanes and in soil-derived branched GDGTs (*br*GDGTs) around the base of the glauconitic silt, which could reflect a tectonically forced regression due to the NAIP thermal uplift. Kaolinite particles are relatively large and heavy and typically deposited closer to the source than finer clays like smectite (Gibbs, 1977; Nilsen et al., 2015). The reduced response in kaolinite deposition to this tectonic event in Denmark compared to the Central North Sea therefore seems

reasonable considering comparably distal position to the NAIP (Figs. 1, 12).

The influx of kaolinite at Fur correlates with a major increase in the CIA and starts minimum 12 cm above the CIE onset, and up to 22 cm (excluding Ash SK2) above the Stolleklint Clay base (Figs. 5, 11) – If we assume the boundary is placed at Ash SK1. Nielsen et al. (2015) found that deposition of kaolinite in the Paleocene–Eocene North Sea thickens substantially towards the Fennoscandian shield, and suggest that this as the main source area

for the Danish sediments. The Fennoscandian shield was characterised by deeply weathered bedrocks in the Paleogene, reflecting the warm tropical Mesozoic climate (Nielsen et al., 2015), and would therefore be enriched in kaolinite. Considering again the typically shorter transport of kaolinite (Gibbs, 1977) and the Danish areas distal position in relation to the NAIP (Figs. 1, 12) it seems likely that the main source of kaolinite was from the Fennoscandian Shield to the north and northeast, despite the main sediment source for the North Sea during this

period being from the west and northwest (Fig. 12; Jordt et al., 2000; Anell et al., 2012). A drop in sea level exposes larger areas to erosion and brings river mouths closer to the marginal marine areas, subsequently triggering an influx of terrestrially derived material that could explain the early Eocene kaolinite and CIA increase in the Danish area. However, the increase post-dates sea level fall and major tectonic uplift in the latest Paleocene, and occurs after the CIE and major temperature increase (Fig. 11). It therefore seems reasonable that it at least partially reflects

an intensified hydrological cycle due to climatic changes rather than purely tectonic forcing. Furthermore, a North Sea surface water freshening is suggested from palynology and shark-tooth apatite $\delta^{18}O$ values in the central North Sea (Zacke et al., 2009; Kender et al., 2012). The observed influx of kaolinite, increased CIA, and rapid intensification of sedimentation rates after the CIE onset at Fur are therefore likely to be the result of increased runoff due to an enhanced hydrological cycle.

**5.2 Volcanic indices**

Although the principal component analysis indicate that the glauconitic silt is most like the Holmehus/Østerrende Formation (Fig. 8), the gradual increase in Ti relative to Fe and K shown by the XRF element core scans suggest a gradual change in lithology towards Ash SK1 (Fig. 6). Variations in the major elements Fe, Ti, and K in marine sediments typically indicate changes in the terrigenous fraction (Rothwell and Croudace, 2015). Titanium is

considered a stable element directly reflecting the coarse-grained terrigenous fraction, and the gradual increase could therefore reflect an influx of terrestrially derived material within the glauconitic silt. This is supported by a slightly coarser grain size, the augmented CIA (Fig. 10), and an influx of terrestrially derived organic matter (Stokke et al., 2020a). However, this influx seems to decline prior to Ash SK1 in contrast to the increase in Ti. Titanium can also be used to indicate volcanic provenance, where K and Ti reflect felsic and mafic sources

respectively (Rothwell and Croudace, 2015). In fact, the K/Ti ratio has been applied as a useful proxy for felsic/mafic provenance in the North Atlantic (Richter et al., 2006). The SK ashes are all highly titanium rich (Figs. 6), and it could be that we see a gradual rise in mafic volcanic activity before the main ash deposition. This is



corroborated by the timing and duration of the East Greenland lava eruptions, which suggest that a 5–6 km thick lava pile was emplaced between 56.0 and 55.6 Ma (Larsen and Tegner, 2006).

An amplified influx of weathered basaltic material such as smectite could also cause the gradually increased Ti flux. However, smectite is already the dominant clay phase in the Holmehus/Østerrende Formation (Fig. 5), and does not show a significant rise in the glauconitic clay. It therefore seems that the augmented Ti concentrations within the glauconitic silt might be caused by an increased ash component, rather than from basaltic weathering. The trace metals Cu, Ni, and V are also found to increase within the glauconitic silt (Fig. 10), all of which are

typically associated with volcanic material and maintain high concentrations within the SK1–SK2 interval. Such volcanic ash deposits that are not visible to the naked eye are called cryptotephras, and typically include glass shards and crystals together with non-volcanic deposits (Cassidy et al., 2014). It is possible that the glauconitic silt includes an increasing portion of cryptotephras prior to the large eruptions producing ashes SK1 and SK2. In our previous study, we found that SSTs cooled prior to the PETM onset in Denmark (Stokke et al., 2020a). Although

the cooling started within the glauconitic silt below Ash SK1 (Fig. 10), we proposed that it could be a result of volcanic cooling induced by $SO_2$ aerosols from NAIP eruptions (Stokke et al., 2020a). The observation of a potentially increasing ash component already within this glauconitic unit could be supportive of a potentially volcanically induced cooling.

The identified tephra layers in Denmark represent explosive eruptions with an unusually large magnitude in order

to be transported such a long distance (Stokke et al., 2020b). The absence of visible tephra layers therefore does not automatically mean that there was no tephra producing eruptions during the PETM. Besides two thin ash layers about 10 and 12 cm above Ash SK2, there are no visible ash layers during the PETM body until the deposition of Ash -39 at about -5 m depth (Fig. 5). However, the box cores allow for a detailed and high resolution overview that might reveal the presence of ash rich intervals earlier. Figure 7 section D shows correlating changes in S, Ca,

Fe/Ti, Fe/K, and K/Ti between about -10.81 to -10.48 m depth. These indicate four possible ash rich horizons within the dark clays, which could be cryptotephras of slightly variable chemistry (Fig. 7 section D). This suggests that explosive volcanic eruptions at a scale substantial enough for some material to reach Denmark may also have occurred during the PETM body. However, much more detailed work is needed in order to confirm the presence or absence of tephra fall deposits during this interval.

**5.3 Changes in basin oxicity**

**5.3.1 Oxic–anoxic shift across the PETM onset**

The PETM CIE is concordant with a lithological shift from the bioturbated Holmehus/Østerrende Formation to the laminated Stolleklint Clay, reflecting a shift to a suboxic to anoxic bottom water environment. An increase in S, pyrite, V/Al, and U (seen as a decrease in Th/U) within the glauconitic silt could indicate reducing conditions prior

to the CIE onset (Fig. 10). The NAIP uplift led to the closing of ocean seaways and North Sea Basin restriction prior to the CIE, resulting in increased halocline stratification that could explain this early deoxygenation (Kender et al., 2012). However, this is contrary to the low organic content, abundant bioturbation, and high content of glauconite, suggesting that an oxygenated environment prevailed pre-PETM. An oxic environment has also been indicated by the relatively high values of the organic biomarker pristane/phytane indicating oxidation of phytol to

pristane (Stokke et al., 2020a). Sluijs et al. (2014) explained a similar co-occurrence of oxic and euxinic proxies





within a section in the Gulf of Mexico as the result of seasonal to decadal variations in basin oxicity. Alternatively, the increase in S, U, and V could be attributed to an increased ash component within the glauconitic silt, as volcanic tephra deposits can reduce the sediment pore waters oxygen levels (Hembury et al., 2012). An increase in the sedimentary ash component has already been suggested based on the high resolution XRF element core scans (Fig.

6), and the increase in ICP-MS analyses of Ni and Cu (Fig. 10). However, the highly redox-sensitive element Mo does not show a similar increase below the CIE (Fig. 10). The burial rate of Mo increases three orders of magnitude in sulfidic environments relative to oxic, as Mo becomes highly reactive in the presence of hydrogen sulfide (Tribovillard et al., 2004; Scott and Lyons, 2012). Molybdenum enrichment above bulk crustal values (1–2 ppm; Taylor and McLennan, 1995) indicates suboxic to anoxic conditions, and enrichments >30 ppm are argued to

indicate euxinic conditions (Scott and Lyons, 2012; Dickson et al., 2014). Molybdenum concentrations at Stolleklint do not increase substantially above 3 ppm until after the CIE, suggesting that a substantial decrease in oxicity does not occur until after the PETM onset.

Laminations occur rapidly after the CIE onset, together with an increase in TOC and Mo concentrations, as well as continued low Th/U values (Fig. 11). The XRF element core scans also show an increase in S and Fe/Ti at base

of the laminated sediments (Fig. 6 section B). Iron and Ti in marine sediments commonly co-vary, and elevated Fe/Ti ratios therefore indicate excess Fe over basaltic lithogenic values (Marsh et al., 2007). While Ti is considered a stable element that directly reflects the coarse-grained terrigenous fraction, Fe is redox-sensitive and may also reflect changes in basin oxicity post-deposition (Rothwell and Croudace, 2015). An increase of Fe relative to Ti or K may therefore indicate suboxic conditions, particularly in concert with increased S content (e.g. Sluijs et al.,

2009). We therefore conclude that the start of laminations about 2 cm above Ash SK2 and the CIE onset indicate the initiation of fully anoxic conditions at Stolleklint. In addition, photic zone euxinia may have occurred just after the CIE onset, as indicated by the presence of sulfur bound isorenieratane; a diagenetic product of green sulfur bacteria (Schoon et al., 2015).

### 5.3.2 Redox and productivity changes during the PETM body

Schoon et al. (2015) argued that photic zone euxinia prevailed during the entire PETM interval in two sedimentary sections in Denmark. Unfortunately, their data from Fur only covers the lowermost 2.5 m and uppermost 0.5 m of the Stolleklint Clay, and therefore excludes most of the PETM body. All proxies from our continuous record through the PETM body suggest that anoxia prevailed throughout (Fig. 10), but the proxies also indicate distinct stratigraphic variations in basin oxicity. The XRF element core scans document a direct correlation between

elevated S and Fe/Ti, and the dark laminations (Fig. 7 section C), suggesting regular fluctuations in basin oxicity (approximately every 2 cm). Variations in basin anoxia is also indicated for longer periods. The biplot of S data from XRF element core scans (Fig. 9) indicate an overall stratigraphic upward increase in basin anoxia. The upward decrease in scatter in the S measurements also indicates that anoxic conditions becomes gradually more continuous with time (Fig. 9).

While Th is sourced from continental weathering and unreactive to redox changes, U has minimal detrital influence and is enriched in the sediments under reducing conditions (Tribovillard et al., 2006). The Th/U ratio therefore reflects U enrichment above crustal values, and can be employed to assess basin oxicity (Wignall and Myers, 1988; Dypvik et al., 2011; Elrick et al., 2017). Th/U is about 4 in the average upper crust, and typically <2 in anoxic environments with substantial authigenic U enrichment (Wignall and Twitchett, 1996). An increase in Th/U above



2 correlates with a decrease in Mo concentrations well below 30 ppm around -17 to -18 m depth in the lower laminated part, indicating a period of less anoxic conditions in the lower PETM body (Fig. 11). Both U and Mo concentrations increase substantially in the upper PETM body, with Th/U <2 and Mo well above 30 ppm indicating euxinic conditions (Fig. 11). Maximum values of S, pyrite and V/Al within this interval also indicate highly reducing conditions, with high TOC of 4 wt% indicating augmented burial rate of organic matter.

The North Sea Basin became very restricted in the latest Paleocene prior to initiation of seafloor spreading and basin subsidence, resulting in poor circulation in the basin (Knox et al., 2010). While this could explain the initial decrease in basin oxicity, there is no further evidence supporting regional uplift and North Sea basin restriction associated with the CIE onset. On the contrary, high HI (Fig. 11), and low input of *br*GDGTs and long-chained *n*-alkanes (Stokke et al., 2020a) suggest that marine-derived organic matter increases up stratigraphy, as the
Stolleklint Clay was likely deposited during a relative sea level rise (Heilmann-Clausen, 1995). Barium is closely related to export productivity, as it precipitates from decaying organic matter (Paytan and Griffith, 2007; Ma et al., 2014). The sedimentary Ba content is at its highest during the upper PETM body, as is $P_2O_5$ (an essential macronutrient), indicating that export productivity was at its highest at this point (Fig. 11; Table 4, Supplement 1). Possible remobilisation of Ba and P needs to be considered, particularly in sulfidic environments (Tribovillard et
al., 2006). However, the dark massive clay in the upper PETM body is also highly enriched in organic matter (TOC up to 3.9 wt%). An increase in TOC could reflect declining terrestrial influx, possibly due to the increasing sea level, which is expected to lead to a decrease in terrestrial sediment transport to marginal areas (e.g. Carmichael et al., 2017). Still, the combined elevation of TOC, Ba/Al, and $P_2O_5$/Al, as well as to some extent Ni/Al and Cu/Al (Fig. 11), are all in support of an increase in export productivity. Kender et al. (2012) found evidence of low surface
water salinity and extensive stratification in the North Sea, and suggested this as the main cause of anoxia during the PETM. A combination of ocean stratification and increased productivity would efficiently contribute to the increase in basin anoxia in the upper PETM body.

**5.4 The PETM recovery**

The PETM carbon cycle perturbations are unusual both in magnitude and duration, and likely a result of a
combination of triggers and feedback mechanisms that are not yet fully understood (McInerney and Wing, 2011; Komar and Zeebe, 2017). Continuous emissions from a light carbon source such as thermogenic degassing around the NAIP could have contributed to the long duration (Svensen et al., 2004; Frieling et al. 2016). Another suggestion is that an initial pulse of light carbon led to warming (e.g. Frieling et al., 2019), subsequently activating positive feedback mechanisms producing continued input of light carbon emissions from sources such as methane
clathrates (Dickens et al. 1995) or terrestrial organic carbon (Bowen 2013). Another key PETM feature is the rapidity of the PETM recovery (e.g. Bowen and Zachos, 2010). The carbon cycle recovery occurs through a combination of natural carbon sequestration and negative feedback mechanisms reducing the atmospheric $CO_2$ content (McInerney and Wing, 2011). Silicate weathering and denudation is perhaps the most important negative feedback mechanisms driving $CO_2$ drawdown (Gislason and Oelkers, 2011), and have been proposed as one of the
most important drivers during the PETM recovery (Kelly et al., 2005; Torfstein et al., 2010; Penman, 2016).

Silicate weathering is highly sensitive to runoff, temperature, and topography (Gislason and Oelkers, 2011). Temperatures rose globally both before and during the PETM onset (Frieling et al., 2017; Frieling et al., 2019). At Stolleklint, temperatures rose at least 10 °C reaching maximum SSTs of ~33 °C shortly after the CIE onset,





followed by a shift to gradually decreasing SSTs throughout the PETM body and recovery (Stokke et al., 2020a).
The warming combined with the increased runoff, indicated in the North Sea by enhanced surface water freshening
(Kender et al., 2012), would result in a warm and humid climate ideal for increased silicate weathering and
denudation. This is supported by the large increase in sedimentation rate, kaolinite influx, and the CIA at Stolleklint
(Figs. 5, 11), suggesting a rapid response in weathering to changes in carbon cycle and temperature. Fresh basaltic
volcanic terrains are particularly prone to weathering, and constitute one of the main sources of weathered
suspended material in the world's oceans (Gislason and Oelkers, 2011). While both the kaolinite content and the
CIA deceases in the upper PETM body, the sedimentation rate likely remained high, suggesting a relatively rapid
influx of other minerals such as the volcanically derived smectite and zeolite. The extensive NAIP flood basalt
volcanism before and during the PETM (e.g. Larsen and Tegner, 2006) may therefore have played an important
role in the enhanced silicate weathering, as reflected in the dominance of smectite within the North Sea (Nilsen et
al., 2015). A second increase in both the CIA and the kaolinite content occur during the CIE recovery at Fur (Figs.
5, 11), as well as further west in the North Sea (Kender et al., 2012; Kemp et al., 2016), supporting an important
role of enhanced silicate weathering in the PETM recovery.

We have documented a relatively rapid rise in silicate weathering as a response to carbon emissions, but a major
increase in productivity and organic carbon burial is delayed until the upper PETM body. An enhanced terrestrial
sediment influx would bring substantial nutrients to the basin, but the decrease in the CIA and kaolinite, as well as
the dominance of marine organic matter (HI>100; Fig. 11), rather suggest a decrease in the terrigenous influx
upwards in the PETM body. However, the deposition of volcanic ash can work as a fertilizer, supplying key
nutrients to the marine environment resulting in augmented productivity (Jones and Gislason, 2008). Substantial
increases in Ba and $P_2O_5$ occur after the deposition of Ash -39 at the end PETM body (Fig. 11; Table 4, Supplement
1). Additional ash deposition below Ash -39 have now been revealed by the possible cryptotephras in XRF element
core scans (Fig. 7 section D), which could have had an added fertilizing effect.

Bowen and Zachos (2010) suggested that the rate of recovery is an order of magnitude faster than expected for
carbon drawdown by silicate weathering alone. Similarly, Penman and Zachos (2018) found that the $\delta^{11}B$ and B/Ca
records of ocean acidification recovers within a similar time frame as the $\delta^{13}C$ record, and far more rapid than
suggested by carbon cycle models that rely on silicate weathering alone (e.g. Zeebe et al., 2009). We have
documented a rise in nutrient supply and enhanced primary production, which could lead to the increased organic
carbon sequestration attributed to the accelerated PETM recovery (Bowen and Zachos, 2010; Komar and Zeebe,
2017; Bridgestock et al., 2019). Enhanced export productivity have also been observed in PETM sites globally
(Bains et al., 2000; Egger et al., 2003; Stein et al., 2006; Soliman et al., 2011; Ma et al., 2014; Bridgestock et al.,
2019), and average Ba burial rates approximately tripled during the PETM (Frieling et al., 2019). Our results show
that negative feedback mechanisms responded rapidly to changes in carbon cycle and SSTs, and remained highly
active from PETM onset to recovery. While the $\delta^{13}C$ values remained low until the PETM recovery, SSTs
decreased gradually throughout the PETM body and recovery. This gradual decline may reflect a temperature
response to the continued carbon drawdown by the alternating increases in both silicate weathering and export
productivity during the PETM.



## Conclusions

We present new mineralogical and geochemical data from an expanded marine section at Fur in northwest Denmark covering the PETM onset, body and recovery. Here, the PETM is defined by a negative 4.5 ‰ CIE and at least 10 °C temperature rise across the PETM onset. The study focuses on a section at Stolleklint, where the
PETM onset is seen as lithological shift from the Holmehus/Østerrende Formation bioturbated clays into the laminated clays of the Stolleklint Clay. The sediments are composed of marine clays, dominated by volcanogenic minerals such as smectite and zeolite, reflecting how important the NAIP was as a source area during this period.

The CIE onset is quickly followed by an increase in kaolinite content, the chemical index of alteration, and substantially enhanced sedimentation rates. This reflects a rapid response in silicate weathering to changes in the
carbon cycle and elevated temperatures, likely due to an enhanced hydrological cycle. Large volumes of easily weathered NAIP flood basalts and widespread tephra deposits likely contributed to accelerate the degree of silicate weathering and carbon drawdown. Proxy evidence shows augmented export productivity and organic matter burial towards the upper PETM body, coinciding with the reappearance of volcanic ash in XRF element core scans and in field exposures. Such a correlation highlights the fertilizing effect of volcanic nutrients, and its potential
importance in increasing primary productivity. Pervasive basin deoxygenation also occurs shortly after the PETM onset, with anoxic to euxinic bottom-water conditions prevailing throughout the PETM body. The continued basin deoxygenation was likely caused by its already restricted nature combined with amplified terrestrial runoff leading to ocean stratification, and intensified export productivity.

The results presented in this study show the potentially rapid environmental response to changes in carbon cycle
and temperature. Our data also show that negative feedback mechanisms were active throughout the PETM. The increased export productivity in the upper PETM body and the renewed rise in kaolinite content and the CIA during the PETM recovery reflect the important role of enhanced silicate weathering and organic matter burial in carbon drawdown leading to the PETM recovery. This highlights the importance of such marginal marine areas in carbon sequestration and recovery from carbon cycle perturbations.

**Data availability**

All of the research data presented in this paper is publically available in the Supplement.

**Supplement**

The supplement related to this article is available online at:

**Author contributions**

EWS, MTJ, and HHS conceptualized and laid out the methodology of the project. EWS, MTJ, LR, HH, IM, BPS, and HHS contributed to data collection and interpretations. MTJ, HH, and HHS contributed with funding acquisition. The original draft was prepared by EWS and MTJ. All authors contributed to the writing in the review and editing stage.



**Competing interests**

The authors declare that they have no conflict of interest

**Acknowledgement**

Claus Heilmann-Clausen, Sverre Planke, Christian Tegner, Tanusha Naidoo, Phil Holdship, Stephen Wyatt, Lina Hedvig Line, and Valentin Zuchuat are all warmly thanked for their assistance. This project was supported by the Research Council of Norway's funding schemes "Unge Forskertallenter" project number 263000 (project
Ashlantic) and "Centres of Excellence" project number 223272.

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
