# Peer review of "Rapid and sustained environmental responses to global warming: The Paleocene–Eocene Thermal Maximum in the eastern North Sea"

_Climate of the Past, 2020_

## Referee Comment (RC1) · Anonymous Referee #1 · 6 Jan 2021

Stokke and co-authors present a detailed multi-proxy record of sediments deposited on Fur, Denmark, that record an expression of the PETM. The authors discuss their data in terms of silicate weathering, basin deoxygenation, volcanic activity and productivity during this interval, and place their findings into a regional (North Sea) and global (carbon cycle processes) context.

The data are excellent and the study is interesting and highly worthy of being published. My comments are for the most part quite minor and I have outlined these line-by-line below. As a broad overview, I would urge the authors to do a few things: i) Standardise their use of elemental data to normalise to Al or enrichment factors. Uncertainties also

need to be stated clearly. ii) Try to edit the text to bring out the main points more clearly. The current manuscript is long and as a reader I felt it lost focus in several places due to the level of detail presented. iii) Please make a better distinction between weathering rate and the character of weathered material. The CIA index (and clay mineralogy) is used as a proxy for a change in the source of weathered material but these data are frequently discussed in terms of 'amount' of weathering, which I found quite off putting. iv) There is quite a bit of repeated information in several of the (many) figures. I wonder if you could consider combining display items?

Line 112: Unit, not section.

Line 132: Be consistent with use of Palaeo... – it is Paleo... in some places.

Line 218: Apostrophe for sediments.

Section 3.4: Note that the use of HI to distinguish OM source relies on low thermal maturity. I would also suggest toning down the ∼100 HI cut off of terrestrial and marine OM – the reality is that these distinctions are much more gradational and subject to mixing etc. The distinction should also lean on OI and Tmax values too.

Section 3.5: ICP-MS (and AES/OES) do not report major elements as oxides, but rather as elemental mass units. Have the authors converted these values to oxide wt equivalent here? Please also note the accuracy (certified RM) and precision (duplicates/RMSE of standards) of the measurements.

Line 268–270: CIA measured 'in situ' reflects the balance of weathering rate (removal of cations, substrate limited) and intensity (rate limited), not rate alone. Also note that marine sediments reflect source mixing – probably more important than hydrodynamic sorting.

Fig. 5: I'm a bit old fashioned but I like to see data points plotted on figures, rather than lines connecting invisible datapoints. This isn't an issue for the clay mineralogy but I would suggest modifying the SST and d13C data displays.

Line 367: There is a shift in interpretation from suboxic to anoxic conditions here for the Stolleklint Clay. The two are different – suboxic places you geochemically in the zone of NO3- / Fe(III) reduction whereas anoxia places you in a zone of sulfate reduction – biogeochemically very distinct. You probably have an oxygenated setting here with subtle – and occasional – shifts to elevated levels of porewater sulfate reduction.

Line 375: I'm not sure about saying 'variability' of the Fe/S data here, rather than the gradient. In fact the gradient changes so much, I wonder if plotting the pyrite Fe/S stoichiometric ratio would be useful to show where you might have 'excess' (non-pyrite) S accumulation.

Line 376: Better to say upsection rather than upwards.

Section 4.4.2: I'm a bit cagey about 'productivity' proxies in the way they are described here. Ba is only useful when present as exported barite, which can become reduced in suboxic porewaters – with the evidence you have for low oxygen conditions, I doubt that Ba/Al is useable in the way employed here and I would suggest removing it. Likewise P/Al is severely biased by the presence of inogranic phosphates that are not uncommon in deposits such as these.

Section 4.4.3: I have a few suggestions here, based around how the various proxies are presented. Firstly, the use of Th/U is a hangover from gamma logging. U reduction is what you are after here, so for consistency, just normalise it to Al and show the UCC ratio for comparison – or better yet, calculate an enrichment factor to compensate for dilution. Second, why isn't Mo normalised to Al like everything else? I suspect this is to enable comparison to the Scott and Lyons (2013) Mo cut-off of 30ppm – but I would suggest removing this. S+L is fine for the modern ocean where we know what the Mo concentration in seawater is, but is sketchy for palaeo-oceans where [Mo] (and other elements) in seawater can vary. Generally a lot of Mo qualitatively = sulfidic conditions and not much Mo = non-sulfidic (or slow sedimentation and adsorption to oxides).

Line 437: Process, not processes.

Line 494: I like this section, but here (and in the results section dealing with clay minerals and CIA) a better distinction could be made better character of detrital material, and amount of detrital material. CIA and mineralogy trace the former, not the latter. Thus, statements such as 'increases in the flux of terrestrially derived material' need to be subtly altered to reflect this distinction.

Line 515: I don't disagree with the role of hydrological cycling as a driver of kaolinite deposition, but what about the transport time lag of source-sink? Sediments do not move instantaneously, so even after a sea-level fall one might expect a lag of several thousand years before seeing a depositional pulse.

Line 564: If you want to discern U enrichment, plot U/Al, not Th/U that requires the reader to invert the axis!

Line 580–583: As is well described in this section, Mo is a sulfidity indicator. So the conclusion that 'oxic' conditions did not deteriorate until after the CIE onset is not supported by these data (lots can happen before Mo starts to react). Th/U actually starts to decline before the CIE onset (Fig. 10) which might support the argument that 'oxic' conditions did deteriorate earlier.

Line 613 (and 701): This is the first mention of 'euxinic.' Better to stick to 'sulfidic' I think, because you don't have any diagnostic evidence here for free $H_2S$ in the water column.

---

## Short Comment (SC1) · 22 Jan 2021

The authors provide an excellent new high-resolution study of a data set spanning the PETM in a sequence that deserves more attention than thus far received. I'd like to express some general concerns related to the referencing in this manuscript.

The introduction and discussion are largely up-to-date with respect to recent advances made on the PETM in general and on the regional expression of the PETM in particular. The last twenty years has seen enormous progress and new approaches to understand a wide array of aspects of the PETM, but many of the underlying observations and ideas emerged the decade before. After the cruise of ODP Leg 113 Maud Rise and

publication of the Kennett and Stott (1991) paper, numerous sections and cores were investigated for the first time with a focus on the PETM. Only a handful of the hundreds of PETM studies published in the 1990s are cited in the present manuscript. Obviously, it is not possible or needed to cover all relevant literature in a research paper. Yet, besides referencing up-to-date research results, in my view it is a good practice to also refer to papers in which certain ideas were first developed and not only to secondary papers further evaluating the same ideas. That good practice is only sparsely applied here.

Referencing can also be inappropriate, for instance when a certain aspect of research results is not even evaluated in a cited paper, but just mentioned in support of other ideas. This applies to line 64: the paper Koch et al. (1992) is referred to with respect to the benthic foraminiferal extinction event. This was a seminal paper, revealing the PETM in continental deposits, but it added nothing to our knowledge of the extinction event. It just briefly mentioned this phenomenon as elaborated by Tjalsma and Lohmann (1982) and Thomas (1990).

With respect to a key topic in this paper, the oceanic anoxia, the following is written in lines 67-71: "Still, globally widespread ocean deoxygenation has been recognised (Pälike et al., 2014; Zhou et al., 2014; Yao et al., 2018), with particularly prevalent anoxic to euxinic conditions observed in semi-enclosed shelf areas such as in the Tethys Ocean (Egger et al., 2003; Khozyem et al., 2013), Peri Tethys Basin (Gavrilov et al., 2003; Dickson et al., 2014), the North Sea (Schoon et al., 2015), and the Arctic Ocean (Stein et al., 2006; Harding et al., 2011)." This seems to suggest that the importance of anoxia in marginal basins during the PETM was only recognized in 2003. Yet, this occurred much earlier, as can be judged from the following two papers: The late Paleocene anoxic event in epicontinental seas of peri-Tethys and formation of the sapropelite unit (Gavrilov et al. 1997- Lithology and Mineral Resources) and Benthic foraminiferal extinction and repopulation in response to latest Paleocene Tethyan anoxia (Speijer et al. 1997 - Geology). (at that time the PETM was still considered as a late Paleocene

event, the LPTM)

Another example is provided by lines 126-128: "Both increased weathering of siliciclastic rocks and enhanced sequestration of organic carbon have been proposed as important negative feedback mechanisms, potentially driving the PETM recovery (Bowen and Zachos, 2010; Ma et al., 2014; Penman, 2016; Dunkley-Jones et al., 2018)." Also the organic carbon sequestration feedback during the PETM was addressed much earlier: Sea-level changes and black shales associated with the late Paleocene thermal maximum: Organic-geochemical and micropaleontologic evidence from the southern Tethyan margin (Egypt-Israel) (Speijer & Wagner 2002 - GSA SP356).

The same applies to lines 82-83: "The 4–5 °C PETM temperature increase (Dunkley Jones et al., 2013; Frieling et al., 2017) is comparable to that predicted in response to the current anthropogenic carbon emissions (e.g. Riahi et al., 2017)." For both statements there are earlier sources addressing them and there are more examples throughout the text for which primary sources are missing.

My general advice is to provide sufficient credit to papers that once brought new ideas and stimuli for further research, next to crediting more recent state-of-the-art papers on the same topic. This should especially apply to the introduction. In this way, the reader can immediately see where and when certain ideas originated and who was behind those ideas.

Further minor suggestions:

Figure 3: Is it correct that the two clay units are not part of a formal lithostratigraphic unit (formation or member)?

Oxicity should be oxygenation.

678: With respect to the discussion on Ba and $P_2O_5$ (export production), it might be worthwhile to consult the PETM records from Israel and Egypt (Schmitz et al. 1997 – Terra nova). This region provided many insights into the PETM, laying the foundation for placing the GSSP for the Ypresian Stage at Dababiya (Dupuis et al. 2003 – Micropaleontology; Aubry et al. 2007 – Episodes).

750: Adatte, T. is correct

790-792: title of SP and editors are missing. Same for 817-818, 834-836, 845-847, 918-920, 921-923, etc.

830, 1039 and 1042: Damsté, J.S.S. should be Sinninghe-Damsté, J.S.

1092-1093: redundant capitals

Recent key papers to also consider: Hollis et al. 2019 (Geoscientific Model Development), Westerhold et al. 2020 (Science)

---

## Referee Comment (RC2) · Anonymous Referee #2 · 9 Mar 2021

Stokke et al. reported new proxy data from Fur, Olst and Store Baelt of Denmark in the North Sea to reconstruct the paleoenvironment of the PETM in this region. This is an overall well-written study with a lot of geochemical data to investigate the environmental changes in response to the warming during the PETM. The paper should be published after the following comments/suggestions are addressed. I list the detailed comments below.

1) Euxinic condition

The authors argue for euxinic condition in the upper half of the PETM body using evidence of Mo (>30 ppm), S (4 wt%), and pyrite (7% of bulk) and low Th/U (<2 ppm).

[Figure]

There is only one data point that shows Mo higher than 30 ppm, which does not correspond with the highest S and a scarce of pyrite data. The authors show that increase in S, pyrite and low Th/U may not be associated with suboxic and anoxic conditions prior to the PETM onset because of bioturbation, and occurrence of glauconite, rather they could be indicators of ash. This suggests that these proxies are not without controversies, and should provide another independent proxy of euxinia. The authors did mentione that previous studies have found biomarker the sulfur bound isorenieratane reported in (Schoon et al., 2015), although that work studied a different site. Therefore, I think the evidence of euxinia is lacking at the study site, because of the lack of data that show consistent > 30 ppm Mo or other independent biomarker data.

2) organic matter burial and export productivity vs. terrestrial organic carbon sequestration

The authors argue that the carbon cycle recovery is aided by increased silicate weathering and export production in the marine realm, rather than the terrestrial carbon sequestration as suggested by Bowen and Zachos (2010). I wonder if the authors could expand their discussion on why they dismiss the regrowth of terrestrial biosphere as a negative feedback mechanism?

3) Calculation of the Chemical Index of Alteration

Did you account for the CaO from the carbonate fraction? You might need the wt% CaCO3 to do that or follow the work of McLennan et al. (1993) to assume reasonable Ca/Na ratios of silicate. Another index that does not require the knowledge of CaO* is CIX (chemical index of alteration without CaO; Garzanti et al., 2014; Harnois, 1988). Please refer to Fedo et al. (1995) paper for the use of CIA with CaO* (which represents Ca in silicate-bearing minerals only), rather than CaO.

Other comments:

1. show the stratigraphic column in Figure 2 with lighologic log and geologic formations

and Period. 2. Line 116: "hundreds of NAIP tephra layers ...": could you provide an age range for these tephra? 3. Line 129: do you mean "organic matter sequestration or burial" rather than "organic matter drawdown"? 4. Section 2. Field area and stratigraphy: Is it possible to provide a paleogeography map of the area? 5. Line 175: How is the CIE magnitude calculated? Please describe the pattern of the CIE, the plateau and recovery in terms of time. 6. Line 179: how is the sedimentation rate calculated? If it has previously published, please briefly describe how it was calculated. 7. Line 502-503: clarify which boundary is placed at Ash SK1, is it the Paleocene-Eocene boundary? 8. Line 539: change Cu, Ni and V to Cu/Al, Ni/Al, and V/Al to reflect what Fig. 10 shows. Also change Al2O3 to Al in Fig. 10 as suggested above. 9. Line 603-604: As previously indicated (Lines 572), the increase in S, U and V could be attributed to an increased ash component with the glauconitic silt, rather than indicating suboxic and anoxic conditions. Can you preclude the contributions from ash to drive up the S values? Same for the argument based on U enrichment below (lines 606-607). 10. Line 626-627: The sentence "An increase in TOC could reflect declining terrestrial influx, possibly due to increasing sea level..." seems lack of support. An increase in TOC could be either due to increase in delivery of terrestrial organic matter, or primary productivity/export productivity of marine organic matter, or increase in preservation due to anoxic conditions. I don't see how an increase in TOC could reflect declining terrestrial influx. It could be that a decreased terrestrial influx along with increased marine primary productivity/export productivity/preservation may lead to an increase in TOC. Is there any evidence for sea level rise in the studied area? If so, a reference is needed for this statement. 11. Line 636: change "light" to "13C-depleted" 12. Line 637: should be "the long duration" of the CIE (or PETM)

Figures Fig. 3. Why is there a gap between Balder Fm. and Horda Fm.?

Fig. 5. Plot the data point rather than showing a line.

Fig. 10. The d13C and SST panel is way too narrow. I think this figure can be separated into two figures to highlight the details of the CIE and temperature. Similar to

Reviewer1, I also suggest plotting raw data points, rather than the smoothed line. The grey colored horizontal bar overlaps with the plot, please change it to another color.

Fig. 11. It is difficult to compare the productivity proxies and redox proxies to the %TOC because their resolution is very different.

Could you provide an image showing the sample preservation in the box core, in addition to the scanning images?

Data It will be helpful to list the analytical data in a table, including Ba (ppm), Al (ppm), etc. Also, why not showing Ba/Al instead of Ba/Al2O3? The productivity proxy is usually by Ba/Al (see Reviewer1 comments), but distinction between terrigenous vs. biogenic barium needs to be made.

---

## Author Comment (AC1) · 12 Apr 2021

We thank the reviewer for the thoughtful comments and suggestions. We have tried to provide a reply to each of the reviewers comments point by point:

i) We did consider using enrichment factors already before the initial submission, but after some deliberation decided on the current presentation in order to be able to compare with certain other studies. Changing to enrichment factors is therefore relatively easy and we have chosen to follow the reviewer's recommendations on this issue. This is also following the recent recommendations from Algeo and Liu (2020), which seems

appropriate. Uncertainties are now stated more clearly in the supplementary material.

ii) We have tried to make each paragraph more succinct and focused to clarify the main points of the paper.

iii) We agree on the importance of this distinction, and we have changed the text both in the methods and discussion to clarify this. It does not alter our interpretation, as our main point is that the rise in the kaolinite and CIA does not reflect an increase in weathering, but rather a change in sediment distribution due to increased precipitation on an already weathered landscape.

iv) We assume the reviewer are referring to the inclusion of the $\delta$13C and SST curves in figures 2, 5, 10, and 11. Figure 2 is included to get an overall overview of the stratigraphy, but we have now combined this figure with figure 5 (now figure 4) to limit the amount of figures. The inclusion of $\delta$13C and SST also in figures 10, and 11 (now 9 and 10) is done in order to make it easier for the reader to place the data in a comparative framework. We do not think it would make it easier for the reader by combining any more of these figures, as they are already relatively data heavy. We are presenting a lot of analytical data combined with a comprehensive site description in this paper, and we therefore feel that this figure setup is necessary to provide clarity for the reader.

Line 112: We have changed this sentence.

Line 132: We believe that we are using the correct nomenclature by using "Paleo" when writing "Paleocene". Paleocene is an epoch that was included much later than the other epochs of the Tertiary/Palaeogene. It was first named not by Charles Lyell, but by the French geologist Wilhelm Philipp Schimper who wrote the name "Paléocène" in French fashion. It means the old Eocene and is an abbreviation of Palaeo and Eocene, therefore written as Pal+Eocene. Were it Palaeo+cene it would mean old recent, which does not make sense. However, if this is an important point for the editor, we will change the wording to Palaeocene.

Line 218: We have changed this sentence.

Section 3.4: As already stated in the result section, the low Tmax values (<422 °C) indicates that the organic matter is immature. We have now stated this more clearly by also citing our previous study (Stokke et al., 2020a) were we show a dominating odd over even preference in long-chained n-alkanes that indicate thermally immature organic matter. It is therefore reasonable to assume that the OM is immature and that we can to some degree use the HI index to indicate OM sources. When that is said, we agree that we can tone down the cut-off and rather focus on the overall variability. We have also now included the OI in the figure, not just in the supplement as before, and have altered the text to take this factor more into consideration. However, we find that these changes does not significantly alter the interpretation that we have an overall increase in marine derived OM up-section. This is also consistent with previous analyses of n-alkanes (Stokke et al., 2020a) where the TAR ratio show an overall upward decrease in terrigenous OM. We have now included the TAR curve in figures 10 and 11 to support the RockEval data.

Section 3.5: The values are converted to oxide wt equivalents, this is now stated in the method section. We have added accuracy and precision in the supplementary material.

Line 268–270: We agree that these are also important factors, and have added the additional definitions to clarify.

Fig. 5: We plotted without data points for simplification, as the points are shown in what is now figure 4. However, it is no problem including them, and we have therefore edited the figure accordingly.

Line 367: We have edited the result section to make it overall less interpretive and as much as possible purely descriptive. The use of these terms are therefore no longer included in this section. We agree that the distinction between suboxic and anoxic is important, and will strive to include this in the discussion. However, we find it difficult

to understand how the reviewer arrive at the conclusion that the environment were partly oxygenated within the lower laminated Stolleklint Clay. While the lamination certainly indicate some regular variation in oxygenation, there are no indication that the environment at any point were actually oxygenated. There is for instance no evidence of bottom-dwelling organisms. On the contrary, there are evidence of green sulfur bacteria, and enrichment of both U and Mo. We therefore do not feel like our discussion should be altered to say it is partly oxygenated in this part of the section, as there just doesn't seem to be any convincing evidence of the fact.

Line 375: We are not entirely sure what the reviewer refer to as the gradient. We are referring to the Biplot of S and Fe counts from the XRF core scans, and try to show that the sediments in the upper dark part of the PETM body have a more homogenous high S concentrations, while lower down in the laminated part the S concentration vary a lot between dark and light laminations. We will try to change the sentence to make this statement clearer.

Line 376: We have changed this sentence.

Section 4.4.2: We agree that these proxies are a bit uncertain, and we will remove them from the paper. '

Section 4.4.3: We thank the reviewer for good suggestions, and have followed the recommendation to switch to enrichment factors. The use of Mo concentrations without normalization was, as guessed, in order to compare with the Scott and Lyon paper. We agree that taking in account the uncertainties in the palaeo-ocean chemistry it may be better to avoid any direct comparison.

Line 437: We have changed this sentence.

Line 494: Again, we agree that this distinction is important, and in line with what we try to communicate. We have altered the text accordingly to clarify the distinction.

Line 515: A potential time-lag is of course relevant, but we still think it is fair to assume

hydrological cycling is the main factor, as this area is not that far from the main kaolinite source area and still it post-dates sea level fall, although of course we don't know the exact timing of the CIE curve. In the Kilda basin (Kender et al., 2012) in the central North Sea we see an increase in kaolinite pre-PETM, most likely due to tectonic uplift of the Scotland-Shetland-Faroe area. Kaolinite is a heavy mineral and not likely to be transported very far. It is therefore likely that the kaolinite at Fur is sourced from the Fennoscandian shield (as also suggested by Nielsen et al., 2015). We therefore think there is ample argument to assume it reflects hydrological cycling and not tectonic activity. Still, source-sink lag is absolutely important, but that would only suggest that the hydrological response to the PETM is even more rapid, as we see the sediment reaction a bit delayed.

Line 564: This will no longer be an issue, as we have switched to enrichment factors.

Line 580–583: We agree that the evidence is not conclusive as to exactly when oxic conditions start to deteriorate, as the influence of ash is complicating the interpretation. However, using enrichment factors, we do see that U is in fact depleted until after the PETM onset. Still, we will note the uncertainty more clearly in the text.

Line 613 (and 701): We have changed the paper to state the uncertainty regarding the presence of euxinic conditions, and conclude with sulfidic.
* * *
Blank

[Figure]

**Fig. 1.** The new figure 4, combining figure 2 and 5.

---

## Author Comment (AC3) · 12 Apr 2021

Firstly, we would like to thank you for reading through our paper so thoroughly and providing constructive feedback. The PETM is a topic of much interest and research, with new papers being published continuously. We agree that it is very important to give credit to the original ideas, and see that we perhaps in some instances have been careless in this regard. Still, this is a research paper, not a review paper, and there is a limit to how many papers are sensible to include. Much has happened in development of both ideas and technology for the last 30 years. Sometimes it will be more relevant for the present study to cite the recent and more advanced research rather than purely

the original ideas. We have still made most of the changes in references you have suggested.

Reply to the minor suggestions:

Figure 3: Yes, that is correct. They are informal units, as also mentioned in the text in section 2.

Line 678: Following the suggestions of the reviewers, we have decided to remove Ba and P2O5 from the paper.

Reference list: Many thanks for your thorough review. We have made corrections in the reference list.

---

## Author Response (AR1)

**Response to reviewers on "Rapid and sustained environmental responses to global warming: The Paleocene–Eocene Thermal Maximum in the eastern North Sea" by Ella W. Stokke et al.**

**Anonymous Referee #1**

Reviewer1: Stokke and co-authors present a detailed multi-proxy record of sediments deposited on Fur, Denmark, that record an expression of the PETM. The authors discuss their data in terms of silicate weathering, basin deoxygenation, volcanic activity and productivity during this interval, and place their findings into a regional (North Sea) and global (carbon cycle processes) context.

The data are excellent and the study is interesting and highly worthy of being published. My comments are for the most part quite minor and I have outlined these line-by-line below. As a broad overview, I would urge the authors to do a few things:

Standardise their use of elemental data to normalise to Al or enrichment factors. Uncertainties also need to be stated clearly.

Response: We did consider using enrichment factors already before the initial submission, but after some deliberation decided on the current presentation in order to be able to compare with certain other studies. Changing to enrichment factors is therefore relatively easy and we have chosen to follow the reviewer's recommendations on this issue. This is also following the recent recommendations from Algeo and Liu (2020), which seems appropriate.

Uncertainties are now stated more clearly in the supplementary material.

Reviewer1: Try to edit the text to bring out the main points more clearly. The current manuscript is long and as a reader I felt it lost focus in several places due to the level of detail presented.

Response: We have tried to make each paragraph more succinct and focused to clarify the main points of the paper.

Reviewer1: Please make a better distinction between weathering rate and the character of weathered material. The CIA index (and clay mineralogy) is used as a proxy for a change in the source of weathered material but these data are frequently discussed in terms of 'amount' of weathering, which I found quite off putting.

Response: We agree on the importance of this distinction, and we have changed the text both in the methods and discussion to clarify this. It does not alter our interpretation, as our main point is that the rise in the kaolinite and CIA does not reflect an increase in weathering, but rather a change in sediment distribution due to increased precipitation on an already weathered landscape.

Reviewer1: There is quite a bit of repeated information in several of the (many) figures. I wonder if you could consider combining display items?

Response: We assume the reviewer are referring to the inclusion of the $\delta^{13}C$ and SST curves in figures 2, 5, 10, and 11. Figure 2 is included to get an overall overview of the stratigraphy, but we have now combined this figure with figure 5 (now figure 4) to limit the amount of figures. The inclusion of $\delta^{13}C$ and SST also in figures 10, and 11 (now 9 and 10) is done in order to make it easier for the reader to place the data in a comparative framework. We do not think it would make it easier for the reader by

combining any more of these figures, as they are already relatively data heavy. We are presenting a lot of analytical data combined with a comprehensive site description in this paper, and we therefore feel that this figure setup is necessary to provide clarity for the reader.

Reviewer1: Line 112: Unit, not section.

Response: We have changed this sentence.

Reviewer1: Line 132: Be consistent with use of Palaeo: : : – it is Paleo: : : in some places.

Response: We believe that we are using the correct nomenclature by using "Paleo" when writing "Paleocene". Paleocene is an epoch that was included much later than the other epochs of the Tertiary/Palaeogene. It was first named not by Charles Lyell, but by the French geologist Wilhelm Philipp Schimper who wrote the name "Paléocène" in French fashion. It means the old Eocene and is an abbreviation of Palaeo and Eocene, therefore written as Pal+Eocene. Were it Palaeo+cene it would mean old recent, which does not make sense. However, if this is an important point for the editor, we will change the wording to Palaeocene.

Reviewer1: Line 218: Apostrophe for sediments.

Response: We have changed this sentence.

Reviewer1: Section 3.4: Note that the use of HI to distinguish OM source relies on low thermal maturity. I would also suggest toning down the 100 HI cut off of terrestrial and marine OM – the reality is that these distinctions are much more gradational and subject to mixing etc. The distinction should also lean on OI and Tmax values too.

Response: As already stated in the result section, the low Tmax values (<422 °C) indicates that the organic matter is immature. We have now stated this more clearly by also citing our previous study (Stokke et al., 2020a) were we show a dominating odd over even preference in long-chained n-alkanes that indicate thermally immature organic matter. It is therefore reasonable to assume that the OM is immature and that we can to some degree use the HI index to indicate OM sources. When that is said, we agree that we can tone down the cut-off and rather focus on the overall variability. We have also now included the OI in the figure, not just in the supplement as before, and have altered the text to take this factor more into consideration.

However, we find that these changes does not significantly alter the interpretation that we have an overall increase in marine derived OM up-section. This is also consistent with previous analyses of n-alkanes (Stokke et al., 2020a) where the TAR ratio show an overall upward decrease in terrigenous OM. We have now included the TAR curve in figures 10 and 11 to support the RockEval data.

Reviewer1: Section 3.5: ICP-MS (and AES/OES) do not report major elements as oxides, but rather as elemental mass units. Have the authors converted these values to oxide wt equivalent here? Please also note the accuracy (certified RM) and precision (duplicates/RMSE of standards) of the measurements.

Response: The values are converted to oxide wt equivalents, this is now stated in the method section. We have added accuracy and precision in the supplementary material.

Reviewer1: Line 268–270: CIA measured 'in situ' reflects the balance of weathering rate (removal of cations, substrate limited) and intensity (rate limited), not rate alone. Also note that marine sediments reflect source mixing – probably more important than hydrodynamic sorting.

Response: We agree that these are also important factors, and have added the additional definitions to clarify.

Reviewer1: Fig. 5: I'm a bit old fashioned but I like to see data points plotted on figures, rather than lines connecting invisible datapoints. This isn't an issue for the clay mineralogy but I would suggest modifying the SST and d13C data displays.

Response: We plotted without data points for simplification, as the points are shown in what is now figure 4. However, it is no problem including them, and we have therefore edited the figure accordingly.

Reviewer1: Line 367: There is a shift in interpretation from suboxic to anoxic conditions here for the Stolleklint Clay. The two are different – suboxic places you geochemically in the zone of NO3- / Fe(III) reduction whereas anoxia places you in a zone of sulfate reduction – biogeochemically very distinct. You probably have an oxygenated setting here with subtle – and occasional – shifts to elevated levels of porewater sulfate reduction.

Response: We have edited the result section to make it overall less interpretive and as much as possible purely descriptive. The use of these terms are therefore no longer included in this section.

We agree that the distinction between suboxic and anoxic is important, and will strive to include this in the discussion. However, we find it difficult to understand how the reviewer arrive at the conclusion that the environment were partly oxygenated within the lower laminated Stolleklint Clay. While the lamination certainly indicate some regular variation in oxygenation, there are no indication that the environment at any point were actually oxygenated. There is for instance no evidence of bottom-dwelling organisms. On the contrary, there are evidence of green sulfur bacteria, and enrichment of both U and Mo. We therefore do not feel like our discussion should be altered to say it is partly oxygenated in this part of the section, as there just doesn't seem to be any convincing evidence of the fact.

Reviewer1: Line 375: I'm not sure about saying 'variability' of the Fe/S data here, rather than the gradient. In fact the gradient changes so much, I wonder if plotting the pyrite Fe/S stoichiometric ratio would be useful to show where you might have 'excess' (non-pyrite) S accumulation.

Response: We are not entirely sure what the reviewer refer to as the gradient. We are referring to the Biplot of S and Fe counts from the XRF core scans, and try to show that the sediments in the upper dark part of the PETM body have a more homogenous high S concentrations, while lower down in the laminated part the S concentration vary a lot between dark and light laminations. We will try to change the sentence to make this statement clearer.

Reviewer1: Line 376: Better to say upsection rather than upwards.

Response: We have changed this sentence.

Reviewer1: Section 4.4.2: I'm a bit cagey about 'productivity' proxies in the way they are described here. Ba is only useful when present as exported barite, which can become reduced in suboxic porewaters –

with the evidence you have for low oxygen conditions, I doubt that Ba/Al is useable in the way employed here and I would suggest removing it. Likewise P/Al is severely biased by the presence of inogranic phosphates that are not uncommon in deposits such as these.

Response: We agree that these proxies are a bit uncertain, and we will remove them from the paper.

Reviewer1: Section 4.4.3: I have a few suggestions here, based around how the various proxies are presented. Firstly, the use of Th/U is a hangover from gamma logging. U reduction is what you are after here, so for consistency, just normalise it to Al and show the UCC ratio for comparison – or better yet, calculate an enrichment factor to compensate for dilution. Second, why isn't Mo normalised to Al like everything else? I suspect this is to enable comparison to the Scott and Lyons (2013) Mo cut-off of 30ppm – but I would suggest removing this. S+L is fine for the modern ocean where we know what the Mo concentration in seawater is, but is sketchy for palaeo-oceans where [Mo] (and other elements) in seawater can vary. Generally a lot of Mo qualitatively = sulfidic conditions and not much Mo = non-sulfidic (or slow sedimentation and adsorption to oxides).

Response: We thank the reviewer for good suggestions, and have followed the recommendation to switch to enrichment factors. The use of Mo concentrations without normalization was, as guessed, in order to compare with the Scott and Lyon paper. We agree that taking in account the uncertainties in the palaeo-ocean chemistry it may be better to avoid any direct comparison.

Reviewer1: Line 437: Process, not processes.

Response: We have changed this sentence.

Reviewer1: Line 494: I like this section, but here (and in the results section dealing with clay minerals and CIA) a better distinction could be made better character of detrital material, and amount of detrital material. CIA and mineralogy trace the former, not the latter. Thus, statements such as 'increases in the flux of terrestrially derived material' need to be subtly altered to reflect this distinction.

Response: Again, we agree that this distinction is important, and in line with what we try to communicate. We have altered the text accordingly to clarify the distinction.

Reviewer1: Line 515: I don't disagree with the role of hydrological cycling as a driver of kaolinite deposition, but what about the transport time lag of source-sink? Sediments do not move instantaneously, so even after a sea-level fall one might expect a lag of several thousand years before seeing a depositional pulse.

Response: A potential time-lag is of course relevant, but we still think it is fair to assume hydrological cycling is the main factor, as this area is not that far from the main kaolinite source area and still it post-dates sea level fall, although of course we don't know the exact timing of the CIE curve. In the Kilda basin (Kender et al., 2012) in the central North Sea we see an increase in kaolinite pre-PETM, most likely due to tectonic uplift of the Scotland-Shetland-Faroe area. Kaolinite is a heavy mineral and not likely to be transported very far. It is therefore likely that the kaolinite at Fur is sourced from the Fennoscandian shield (as also suggested by Nielsen et al., 2015). We therefore think there is ample argument to assume it reflects hydrological cycling and not tectonic activity. Still, source-sink lag is absolutely important, but that would only suggest that the hydrological response to the PETM is even more rapid, as we see the sediment reaction a bit delayed.

Reviewer1: Line 564: If you want to discern U enrichment, plot U/Al, not Th/U that requires the reader to invert the axis!

Response: This will no longer be an issue, as we have switched to enrichment factors.

Reviewer1: Line 580–583: As is well described in this section, Mo is a sulfidity indicator. So the conclusion that 'oxic' conditions did not deteriorate until after the CIE onset is not supported by these data (lots can happen before Mo starts to react). Th/U actually starts to decline before the CIE onset (Fig. 10) which might support the argument that 'oxic' conditions did deteriorate earlier.

Response: We agree that the evidence is not conclusive as to exactly when oxic conditions start to deteriorate, as the influence of ash is complicating the interpretation. However, using enrichment factors, we do see that U is in fact depleted until after the PETM onset. Still, we will note the uncertainty more clearly in the text.

Reviewer1: Line 613 (and 701): This is the first mention of 'euxinic.' Better to stick to 'sulfidic' I think, because you don't have any diagnostic evidence here for free H2S in the water column.

Response: We have changed the paper to state the uncertainty regarding the presence of euxinic conditions, and conclude with sulfidic.

**Anonymous Referee #2**

Reviewer2: Stokke et al. reported new proxy data from Fur, Olst and Store Baelt of Denmark in the North Sea to reconstruct the paleoenvironment of the PETM in this region. This is an overall well-written study with a lot of geochemical data to investigate the environmental changes in response to the warming during the PETM. The paper should be published after the following comments/suggestions are addressed. I list the detailed comments below.

Response: We would firstly like to clarify that we do not show any data from Ølst and Storebelt, as also described in the field area and sampling sections. We assume this misunderstanding is based on the fact that we included these localities in figure 1. This was done to help the reader understand the local geography as these localities are mentioned in the text due to other important studies having focused on these places. We have now altered figure 1 and hope that will deter any confusion in the future.

Reviewer2: 1) Euxinic condition

The authors argue for euxinic condition in the upper half of the PETM body using evidence of Mo (>30 ppm), S (4 wt%), and pyrite (7% of bulk) and low Th/U (<2 ppm). There is only one data point that shows Mo higher than 30 ppm, which does not correspond with the highest S and a scarce of pyrite data. The authors show that increase in S, pyrite and low Th/U may not be associated with suboxic and anoxic conditions prior to the PETM onset because of bioturbation, and occurrence of glauconite, rather they could be indicators of ash. This suggests that these proxies are not without controversies, and should provide another independent proxy of euxinia. The authors did mention that previous studies have found biomarker the sulfur bound isorenieratane reported in (Schoon et al., 2015), although that work studied a different site. Therefore, I think the evidence of euxinia is lacking at the study site, because of the lack of data that show consistent > 30 ppm Mo or other independent biomarker data.

Response: We would argue that there is not just 1, but 6 datapoints where Mo is > 30, making the entire upper half of the PETM very high in Mo, and both S and pyrite is declining in the uppermost 4 of these. As a result of the review's suggestions, we have chosen to switch to enrichment factors and subsequently not compare with the 30 ppm cutoff due to issues with comparison with modern seawater.

This does however, not change the fact that the upper half of the PETM is very high on both Mo and U. These elements are not increasing in the ash-rich section pre-PETM. On the contrary, $U_{EF}$ is rather depleted at these intervals. There are also overall no known great connection between ash content and high sedimentary U, Mo, or indeed S content. There is therefore not reason to believe that the increase in these during the PETM is related to volcanic ash. Especially as there is limited evidence for extensive ash deposition during the PETM body. Regarding the study of Schoon et al. (2015), their data is from the exact same location, as they present data both from Stolleklint and Store Bælt. There is therefore in fact found sulfur bound isorenieratane in the exact same section as we are working. We will change the text to avoid any confusion on this point.

Although no proxies are without complications, we believe that the combination of so many proxies suggesting lowered oxygen conditions during the PETM is fairly good evidence of this fact. When that is said, we will based on both reviewers comments refrain from using the term euxinic, and rather stay with anoxic and sulfidic.

Reviewer2: 2) Organic matter burial and export productivity vs. terrestrial organic carbon sequestration

The authors argue that the carbon cycle recovery is aided by increased silicate weathering and export production in the marine realm, rather than the terrestrial carbon sequestration as suggested by Bowen and Zachos (2010). I wonder if the authors could expand their discussion on why they dismiss the regrowth of terrestrial biosphere as a negative feedback mechanism?

Response: We do not intentionally dismiss this theory, but we have not focused particularly on it. Our data show an increase in marine OM up section, which does not lend support to the importance of growth of terrestrial biosphere. However, it does not exclude it neither, as there are many factors controlling the relative distribution of terrestrial and marine OM such as changes in sea level, and we know for a fact that the sea-level rose. There are many theories regarding the termination of the PETM, and we cannot in the scope of our paper go into a detailed discussion of all. As our data does not really support nor exclude the regrowth of terrestrial biosphere as an important negative feedback mechanism, we believe greater focus on this hypothesis is beyond the scope of this paper. Yet, we do cite the paper, and have now mentioned the theory more specifically. Our data is from a marine section, and we suggest using other sections to test the strength of Bowen and Zachos (2010).

Reviewer2: 3) Calculation of the Chemical Index of Alteration

Did you account for the CaO from the carbonate fraction? You might need the wt% CaCO3 to do that or follow the work of McLennan et al. (1993) to assume reasonable Ca/Na ratios of silicate. Another index that does not require the knowledge of CaO* is CIX (chemical index of alteration without CaO; Garzanti et al., 2014; Harnois, 1988). Please refer to Fedo et al. (1995) paper for the use of CIA with CaO* (which represents Ca in silicate-bearing minerals only), rather than CaO.

Response: We had not taken carbonate fraction into account, as this is minimal in the analyzed sediments. However, we agree that it is better practice to do so, and following the reviewers recommendations we have now included this. Still, it does not significantly alter the results and has no impact on the interpretation.

Reviewer2: 1. show the stratigraphic column in Figure 2 with lighologic log and geologic formations and Period.

Response: Based on comments of reviewer 1, we have combined figures 2 and 5 to limit the amount of figures. We have also altered what is now figure 4 to include the period and epoch.

Reviewer2: 2. Line 116: "hundreds of NAIP tephra layers : : :": could you provide an age range for these tephra?

Response: In the introduction, we refer to the overall ash production, not just the ashes included in the part of the section we are working. The age range for the full Danish ash series is noted in figure 3 (now fig. 2) and for the studies section it is noted in figure 5 (now fig. 4). We have now also stated the age range more clearly in the text and added a reference to figure 3.

Reviewer2: 3. Line 129: do you mean "organic matter sequestration or burial" rather than "organic matter drawdown"?

Response: We have changed this sentence.

Reviewer2: 4. Section 2. Field area and stratigraphy: Is it possible to provide a paleogeography map of the area?

Response: We have palaeogeographic maps of the area both in figures 1 and 12 (now fig. 11).

Reviewer2: 5. Line 175: How is the CIE magnitude calculated? Please describe the pattern of the CIE, the plateau and recovery in terms of time.

Response: It has not been possible to perform any astronomical calibration of the PETM CIE at Fur, and we therefore cannot describe the pattern in terms of time beside what is already assumed as the general timing of the PETM. We have now stated this in the text, and we refer to figure 5 (now fig. 4) where we have noted what timings we know.

Reviewer2: 6. Line 179: how is the sedimentation rate calculated? If it has previously published, please briefly describe how it was calculated.

Response: We have included a brief explanation of this in the text.

Reviewer2: 7. Line 502-503: clarify which boundary is placed at Ash SK1, is it the Paleocene-Eocene boundary?

Response: We are referring to the lithological boundary between the Stolleklint Clay and the underlying Holmehus/Østerrende Fm. The exact position of this is a bit uncertain, but based on the PCA of the XRF scanning data we assume it is placed here. We have changed the text to clarify.

Reviewer2: 8. Line 539: change Cu, Ni and V to Cu/Al, Ni/Al, and V/Al to reflect what Fig. 10 shows. Also change Al2O3 to Al in Fig. 10 as suggested above.

Response: We have now changed to enrichment factors, so this is no longer an issue.

Reviewer2: 9. Line 603-604: As previously indicated (Lines 572), the increase in S, U and V could be attributed to an increased ash component with the glauconitic silt, rather than indicating suboxic and anoxic conditions. Can you preclude the contributions from ash to drive up the S values? Same for the argument based on U enrichment below (lines 606-607).

Response: The reason we suggest that the increase in particularly V and S below ash SK1 could be due to a high ash component, is because there are ample chemical evidence of a high degree of ash within these few cm of sediment. This we have from the XRF core scans, the ICP-MS analyses, and not least from the fact that there is deposition of two thick ash layers just above. That means that we may be analyzing just as much ash as sediment in our bulk-rock analyses.

This is not the case within the upper PETM body. While there may be some indication of potential thin cryptotephras, this evidence is limited. There are also no major ash layers deposited, and no evidence of bioturbation that could redistribute ash into the sediments. There is also not really very well known that volcanic derived material is particularly rich in U. On the contrary, U is known to be a redox sensitive element with little detrital influence (e.g. Tribovillard et al., 2006). Combined with the high TOC content and Mo enrichment, as well as the sedimentological evidence that the clay is almost black, it is ample evidence that the sediments are likely to be highly anoxic in this part of the section. If ash plays a role in elemental enrichment here, it is comparatively negligible.

Reviewer2: 10. Line 626-627: The sentence "An increase in TOC could reflect declining terrestrial influx, possibly due to increasing sea level: : :" seems lack of support. An increase in TOC could be either due to increase in delivery of terrestrial organic matter, or primary productivity/export productivity of marine organic matter, or increase in preservation due to anoxic conditions. I don't see how an increase in TOC could reflect declining terrestrial influx. It could be that a decreased terrestrial influx along with increased marine primary productivity/export productivity/preservation may lead to an increase in TOC. Is there any evidence for sea level rise in the studied area? If so, a reference is needed for this statement.

Response: We refer here to the fact that sometimes a decrease in detrital input can give the illusion that there is an enhanced preservation of OM, as suggested in Carmichael et al. (2017) that we also refer to in the text. However, we do also reject this theory as a possible cause in the paper. In the revised version, we have omitted this sentence, as it really is of minor importance.

Reviewer2: 11. Line 636: change "light" to "13C-depleted"

Response: We have changed this sentence.

Reviewer2: 12. Line 637: should be "the long duration" of the CIE (or PETM)

Response: We have changed this sentence.

Reviewer2: Figures Fig. 3. Why is there a gap between Balder Fm. and Horda Fm.?

Response: The gaps in the stratigraphy refer periods of no deposition or erosion, so-called hiatuses. The stratigraphy in the figure is based on King (2016) and Schiøler et al. (2007), which can provide a more thorough explanation of the central North Sea stratigraphy.

Reviewer2: Fig. 5. Plot the data point rather than showing a line.

Response: We have now plotted data points.

Reviewer2: Fig. 10. The d13C and SST panel is way too narrow. I think this figure can be separated into two figures to highlight the details of the CIE and temperature. Similar to Reviewer1, I also suggest plotting raw data points, rather than the smoothed line. The grey colored horizontal bar overlaps with the plot, please change it to another color.

Response: The d13C and SST data is shown in greater detail in figure 2 (now figure 4), and were mainly included in figures 10 and 11 for comparative purposes. We have now changed the figures to have a slightly wider panel and plotted the data points. We do not agree that it would be appropriate to separate the figure into 2 figures, as it is easier to compare the data when plotted together. They grey horizontal lines represent ash layers and are supposed to cut the vertical plot lines. We have made them darker to avoid confusion.

Reviewer2: Fig. 11. It is difficult to compare the productivity proxies and redox proxies to the %TOC because their resolution is very different.

Response: Yes, TOC and trace metals are normally given with different scales. We are not sure what the reviewer wishes us to do about that.

Reviewer2: Could you provide an image showing the sample preservation in the box core, in addition to the scanning images?

Response: We do not believe it would be constructive to add yet another image of the box-cores as there are already two of each. We refer the reviewer to figures 6 and 7 (now 5 and 6) for pictures of the box cores. A note on sample preservation has been added to the figure text. The sample preservation of the box cores is good, as the sediments were overall soft and easy to sample. There were some cracks in the cores covering the onset that are evident in the images already provided. These were avoided during XRF scanning, which was conducted on a smoothed plane surface along the middle.

Reviewer2: Data It will be helpful to list the analytical data in a table, including Ba (ppm), Al (ppm), etc. Also, why not showing Ba/Al instead of Ba/Al2O3? The productivity proxy is usually by Ba/Al (see Reviewer1 comments), but distinction between terrigenous vs. biogenic barium needs to be made.

Response: Based on the reviewer's comments, we have decided to remove the productivity proxies Ba and P2O5. We do not have sufficient data to distinguish between terrigenous vs biogenic barium, and as there seems to be extensive uncertainty in using whole-rock sedimentary Ba as a productivity proxy we decide that the safest option is to omit this from the study.

---

## Author Response (AR2)

**Response to reviewer and editor on the manuscript:**

"Rapid and sustained environmental responses to global warming: The Paleocene–Eocene Thermal Maximum in the eastern North Sea"

***From the editor:***

Dear

Thanks for the revised version. However, you would need to clarify one issue that the referee raised:

"It appears that the chemical index of alteration has been recalculated, but I'm not sure if it is reflected in Fig. 9 and Fig. 10."

Thanks in advance.

With the best wishes

Zhengtang Guo

***Our reply:***

We understand that this may be a bit unclear. In the last revision, reviewer 2 asked us to recalculate the Chemical Index of Alteration (CIA) taking in account the carbonate fraction of the sediments. We had initially not done this, as we knew the carbonate fraction in the sediments was minimal. However, following this recommendation, we recalculated considering both the fraction of carbonate and apatite, as is often done with the CIA. However, as also stated in our last response to reviewers, this led to a minimal change in the CIA, and no change in the interpretation. However, we should maybe have been clear on how small this change was. It is the order of 0.4-3.2 %, and will therefore not be very visible on a graph plotted between 60-90%. Consequently, it looks like the graphs are not altered. To be absolutely certain, we have remade the figures with the correct graph and uploaded a new manuscript with new versions of Figures 9 and 10. We hope that this will clarify the issue.